# Probabilistic Tsunami Hazard Analysis for Tuzla Test Site Using Monte Carlo Simulations

H. Basak Bayraktar[1,2,3], Ceren Ozer Sozdinler[4]

[1]Department of Geophysics, Kandilli Observatory and Earthquake Research Institute, Bogazici University, Istanbul, 34684, Turkey
[2]Department of Physics "Ettore Pancini", University of Naples Federico II, Naples, 80126, Italy
[3]Istituto Nazionale di Geofisica e Vulcanologia, Rome, Italy
[4]Institute of Education, Research and Regional Cooperation for Crisis Management Shikoku, Kagawa University, Takamatsu, 760-8521, Japan

*Correspondence to*: Hafize Başak Bayraktar (hafizebasak.bayraktar@unina.it)

**Abstract.** In this study, time-dependent probabilistic tsunami hazard analysis (PTHA) is performed for Tuzla, Istanbul in the Sea of Marmara, Turkey, using various earthquake scenarios of Prince Island Fault (PIF) within next 50 and 100 years. Monte Carlo (MC) simulation technique is used to generate a synthetic earthquake catalogue, which includes earthquakes having moment magnitudes between $M_w$ 6.5 and 7.1. This interval defines the minimum and maximum magnitudes for the fault in the case of entire fault rupture, which depends on the characteristic fault model. Based on this catalogue, probability of occurrence and associated tsunami wave heights are calculated for each event. The study associates the probabilistic approach with tsunami numerical modelling. Tsunami numerical code NAMI DANCE was used for tsunami simulations. According to the results of the analysis, distribution of probability of occurrence corresponding to tsunami hydrodynamic parameters are represented. Maximum positive and negative wave amplitudes show that tsunami wave heights up to 1 m have 65% probability of exceedance for next 50 years and this value increases by 85% in Tuzla region for the next 100 years. Inundation depth also exceeds 1m in the region with probabilities of occurrence of 60% and 80% for the next 50 and 100 years, respectively. Moreover, probabilistic inundations maps are generated to investigate inundated zones and the amount of water penetrated inland. Probability of exceedance of 0.3 m wave height ranges between 10% and 75% according to these probabilistic inundation maps and the maximum inundation distance calculated among entire earthquake catalogue is 60 m in this test site. Furthermore, synthetic gauge points are selected along the western coast of the Istanbul by including Tuzla coasts. Tuzla is one of the areas that show high probability exceedance of 0.3 m wave height, which is around 90%, for the next 50 years while this probability reaches up to more than 95% for the next 100 years.

## 1 Introduction

Marmara Region, especially highly populated cities along the coasts of the Marmara Sea, is the heart of Turkish economy in terms of having great number of industrial facilities in largest capacity and potential, refineries, ports and harbors. The Marmara Sea and the area is one of the most seismically active areas in Turkey. Main active faults of the region pass through the

Marmara Sea. Thus, coastal cities in Marmara region, especially Istanbul, which has significant importance in terms of economy, and historical and sociocultural heritage with a population of more than 15 million, is under the threat of high damage due to possible big earthquake and also triggered tsunamis. Recent studies and evaluation of earthquake recurrence periods

revealed that there is a high possibility of having an earthquake with magnitude larger than $M_w$ 7.0 in PIF. According to Ambraseys (2002), the latest earthquake on this fault system occurred in 1766 and since that time, this fault has been accumulating huge amount of energy. According to Parsons (2004), the probability of occurrence of M>7 earthquake beneath the Marmara Sea was estimated to be 35-70% in the following 30 years. The region has distinctive characteristics in terms of its complex tectonic structure and high possibility of an earthquake occurrence with the magnitude larger than 7.0 offshore

Istanbul mega-city. Therefore, there has been a wide range of studies in Marmara Sea region regarding the fault mechanisms, seismic activities, earthquakes and triggered tsunamis (Armijo et al., 2002; Armijo et al., 2005; Okay et al., 1999; Le Pichon et al., 2001; Yaltirak 2002; McNeill et al., 2004; Aksu et al., 2000; Imren et al., 2001; Pondard et al., 2007, Yalçıner et al., 1999; Yalçıner et al., 2000; Yalçıner et al., 2002; Aytore et al., 2016 ; Hebert et al., 2005; Altınok et al., 2001; Altınok et al., 2003; Guler et al., 2015; Cankaya et al., 2016; Tufekci et al., 2018; Latcharote et al., 2016 ).

The North Anatolian Fault Zone (NAFZ) controls the great part of the seismic activity in the Marmara Sea region. The fault zone sets apart Anatolia (Asian part of Turkey) and Eurasia due to the northward migration of Arabian Plate in the east and southward rollback of the Hellenic subduction zone in the west as seen in Fig. 1 (Armijo et al., 1999; Flerit et al., 2004; Le Pichon et al., 2015).

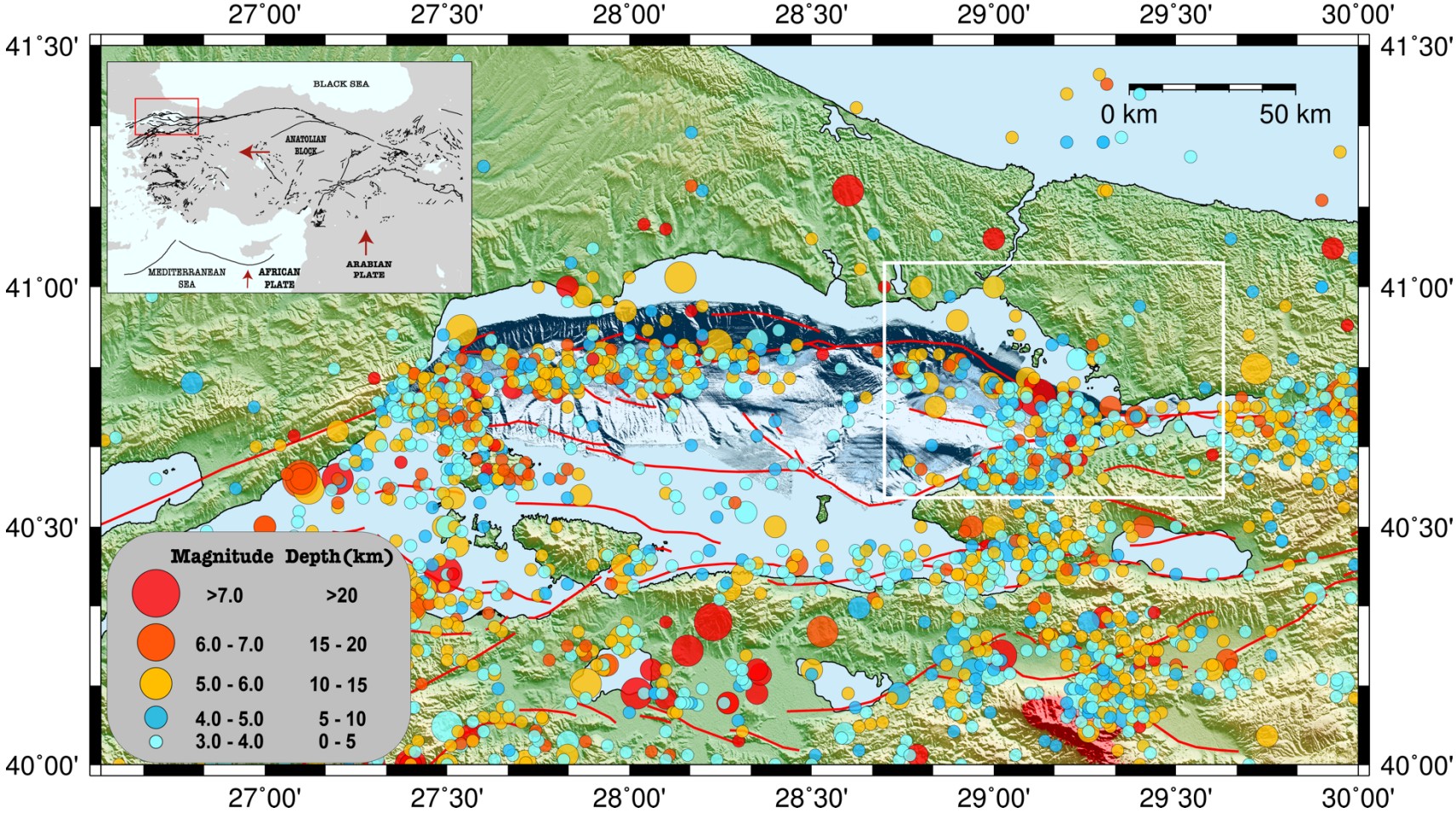

Figure 1: Seismicity map of the Marmara region and general tectonic map of the Turkey on the top – left. In the seismicity map, the size of the circle's changes with magnitude of the earthquakes and the color of the circles defines the depth change of the earthquakes. Red lines show the known active faults (Modified from Emre et al., 2013) in the region and white square is the area with the PIF. In the general tectonic map of Turkey, red arrows show the direction of the plate motion, black lines show the active faults in the region (Modified from Emre et al., 2013) and red rectangular shows the Marmara region (created using The Generic Mapping Tools, Version 5.4.1).

The Marmara Sea region is a transition zone between the strike-slip regime of the NAFZ and the extension regime of the Aegean Sea area (on the top – left of Fig 1). The northern branch of the NAFZ forms a major transtensional NW- SE right bend under the Sea of Marmara at Çınarcık trough (Murru et al., 2016). The fault trace is attached to the complex Central Marmara and Tekirdağ pull-apart basins, before joining the NE-SW striking Ganos fault on land by following the northern margin of the Marmara Sea. Finally, the fault exits into the Aegean Sea by way of Saros Gulf (Wong et al., 1995; Armijo et

al., 1999; Armijo et al., 2002; Okay et al., 1999; Le Pichon et al., 2001; Yaltirak 2002; McNeill et al., 2004). The fault trace beneath the Marmara Sea is not directly observable. Therefore, making a segmentation model for the offshore parts of the NAFZ is quite challenging, which causes the fault dimensions, such as its length and width, to include a sum of error margin (Aksu et al., 2000; Imren et al., 2001; Le Pichon et al., 2001; Armijo et al., 2002; Armijo et al., 2005; Pondard et al., 2007). The current right-lateral slip rate along the NAFZ is about 25 mm/yr (Meade et al., 2002; Reilinger et al., 2006). In the western

side, the motion between the Anatolia and Eurasia plates is accommodated across the Marmara region by ~ 19 mm/yr of right-lateral slip and 8 mm/yr of extension (Flerit et al., 2003; Flerit et al., 2004). Slip rates of the main Marmara fault ranges between 17-28 mm/yr (Le Pichon et al., 2003; Reilinger et al., 2006). On the other hand, Hergert and Heidbach (2010) suggests that the right-lateral slip rate on the main Marmara fault is between 12.8-17.8 mm/yr due to slip partitioning and internal deformation. The right-lateral slip rate for the PIF and Çınarcık basin is 15±2 mm/yr and in addition to this, the fault has 6±2 mm/yr of

extension (Ergintav et al., 2014).

The main characteristic of the NAFZ is having earthquakes systematically propagated westward and historical records show that, northern strand of the NAFZ generates an earthquake with the recurrence interval of about 250 years beneath the Marmara Sea and the latest event occurred in 1766 (Ambraseys, 2002, Bohnhoff et al., 2013). This event caused the rupture of the 58 km long northern part of NAFZ from Izmit to Tekirdağ (Ambraseys and Finkel, 1995; Ambraseys and Jackson, 2000).

However, the earthquake that happened on 2 September 1754 can be considered as the latest characteristic event for the PIF segment and it caused the rupture of a 36 km long fault segment (Ambraseys and Jackson, 2000). The NAFZ has experienced two M>7 earthquakes in August 1912 Ganos and August 1999 Izmit earthquakes recently. After the 1999 Izmit event, seismic energy along the 150 km long northern part of the NAFZ has been accumulating continuously since 22 May 1766 earthquake. This fault zone extends right next to south of Istanbul beneath the Marmara Sea, and this situation increases the rupture

possibility of the PIF and the risk for megacity Istanbul (Stein et al., 1997; Barka 1999; Bohnhoff et al., 2013). Ergintav et al., (2014) also indicated that the PIF segment accumulates stress 15±2 mm/yr and the 3.7m slip deficit has been accumulating since the 1766 events and this makes PIF most likely to generate the next M > 7 earthquake along the Sea of Marmara segment of the NAF.

Beside these seismic activities in the region, studies on the historical tsunami records show that 35 tsunami events happened

between BC 330 and 1999 in the Marmara Sea region and the majority of them are earthquake-related tsunami events (Altinok et al., 2011; Yalçıner et al., 2002). The 1509 earthquake, with an estimated magnitude around 8.0, is one of the examples for these events. This earthquake triggered a tsunami and the tsunami waves inundated along Istanbul coasts, reaching the city walls and around 4000–5000 people died in the city (Ambraseys and Finkel, 1995). The 1894 earthquake is also one of the

important events that happened in the Marmara Sea. The earthquake triggered a tsunami and the sea inundated 200 m in Istanbul (Altinok et al., 2011). The recent one happened after the 17 August 1999 Izmit earthquake and after the earthquake, E-W trending tectonic deformation along the basin and submarine failures generated a tsunami. The International Tsunami Survey Team (Yalçıner et al., 1999; Yalçıner et al., 2000) investigated the region and they observed 2.66 m run-up along the coast from Tütünçiftlik to Hereke and 2.9 m run-up at Değirmendere (Yalçıner et al., 2002).

Several tsunami hazard estimation studies (Ozer Sozdinler et al., 2019; Hancilar, 2012; Aytore et al., 2016; Hebert et al., 2005) were also conducted in the region. These tsunami analyses were mostly performed in deterministic manner using various earthquake scenarios depending on the combinations of different fault parameters without considering probability of occurrences. The 40 km long fault in Eastern Basin of Marmara Sea, with a significant normal component, may generate tsunami wave which can reach maximum 2 m height along the Istanbul coast with locally considerable inundation (Hebert et al., 2005). The rupture of Yalova Fault, PIF or Central Marmara Fault can also cause a serious damage along the coast of Istanbul. Tsunami wave heights can reach 4.8m and can penetrate 340m inwards from the coast in Haydarpaşa Port (Aytore et al., 2016).

A few probabilistic seismic and tsunami hazard analyses (Murru et al., 2016; Erdik et al., 2004; Hancilar, 2012) were also done in this region. Seismic hazard maps were prepared in the Marmara Sea region by describing fault segments and peak ground accelerations with the periods corresponding to 10% and 2% probabilities of exceedance in 50 years (Erdik et al., 2004). Besides that, tsunami inundation maps are prepared based on probabilistic and deterministic analyses by depending on these segmentations (Hancilar, 2012). Time-dependent and time-independent earthquake ruptures are also estimated in the Marmara Sea region for the next 30 years (Murru et al., 2016). These previous studies have been conducted for entire Marmara Sea region and therefore they give general and rough information about probability of occurrence in the region without focusing on any specific region in high resolution. However, probabilistic tsunami hazard assessment is important to calculate the tsunami exposure and risk on human populations and infrastructures, since probability calculations consider all possible earthquakes in a fault even if they occur with very low probability (Løvholt et al., 2012; Løvholt et al., 2015; Grezio et al., 2017). The results of probabilistic studies should be considered when decision makers design coastal zones and structures, especially critical ones. Different from previous probabilistic approaches in the Marmara Sea, the probability of earthquake occurrences in one fault segment, PIF, are taken into account for the preparation of high-resolution tsunami inundation maps and distribution of hydrodynamic parameters due to the probability of occurrence of associated earthquakes on PIF determined by MC Simulations.

This PTHA study depends on the fully characteristic fault model and the main purpose is to perform PTHA for selected test site. Tuzla test site is one of the coastal districts of Istanbul and located on the southernmost part of the city (Fig 2). The region includes several residential areas, but the most critical point about the region is that Tuzla has the biggest shipyard area not only in the Marmara Sea but also in Turkey (Fig 3). In this study we mainly focused on this region because it is about 20km away from the PIF and therefore has high risk of both earthquake and tsunami damage.

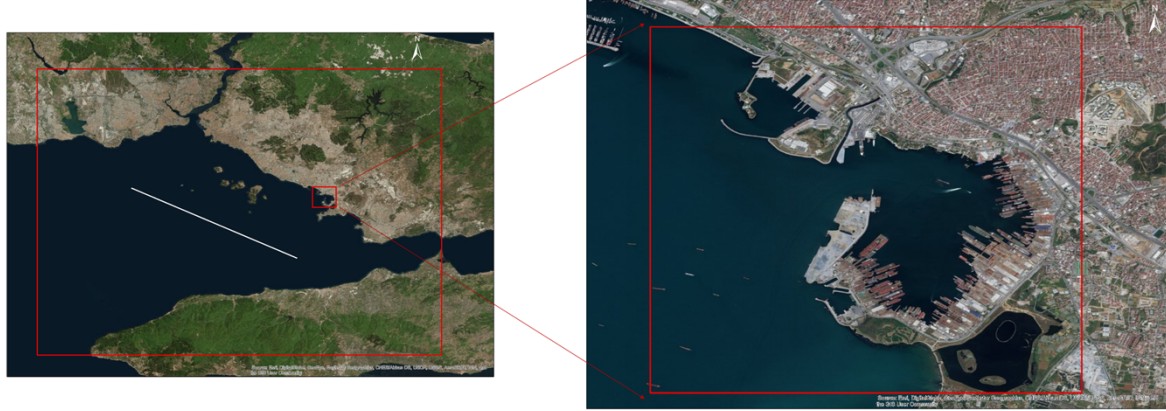

**Figure 2: The Marmara Sea Region, Tuzla Test Site and the Location of PIF Segment which is used in the analysis like a straight line (created using ArcMap Version 10.5).**

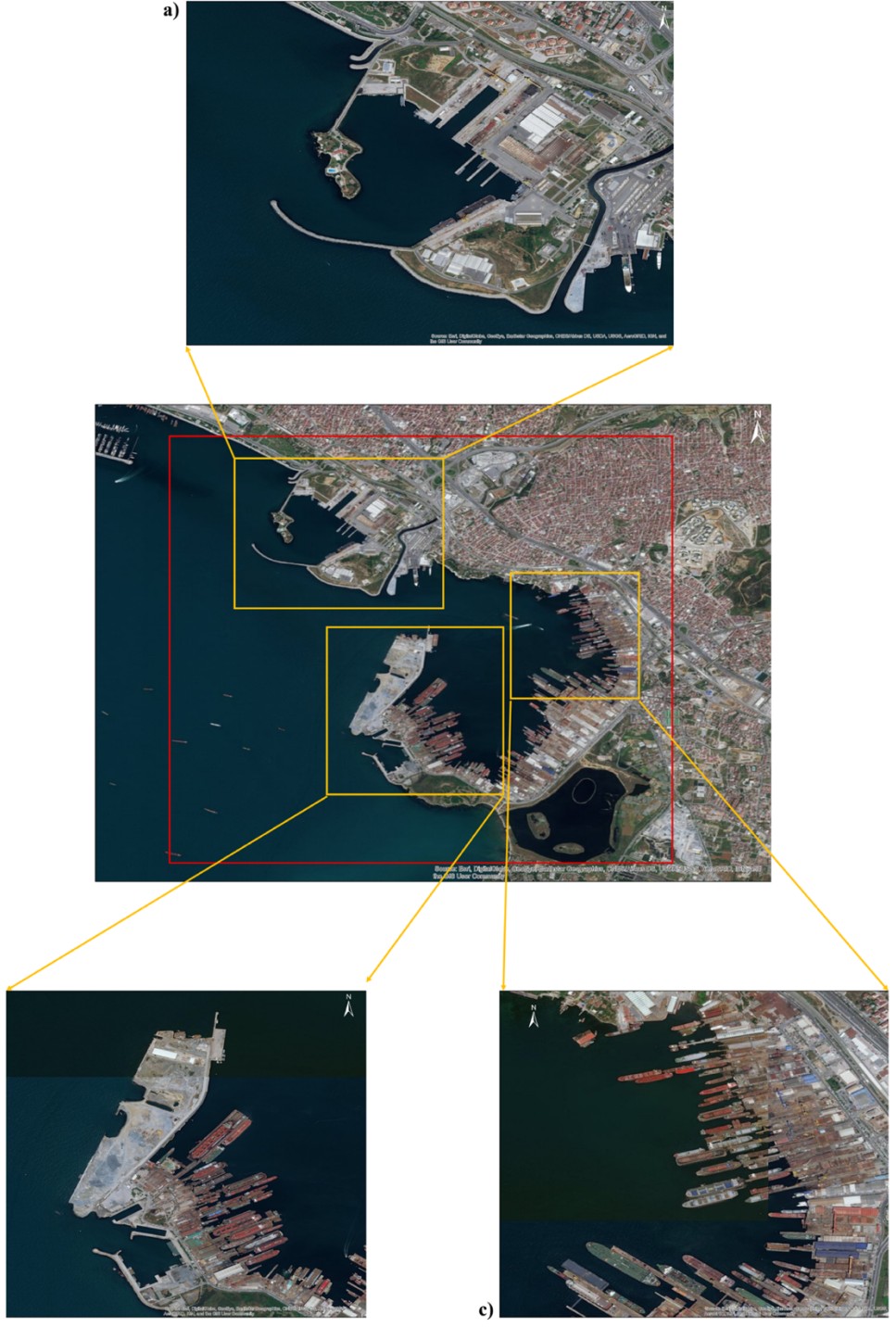


**Figure 3: Some Important Locations at Tuzla Domain (a)Northern part of the Tuzla domain. (b) Southern part of the Tuzla Domain. (c) Tuzla Shipyard (created using ArcMap Version 10.5).**

## 2 Probabilistic Analysis

Probabilistic tsunami hazard analysis (PTHA), as it is recently becoming a widely-used procedure for coastal zones, is
performed for Tuzla region, Istanbul. This method has been applied for various tsunami sources, such as earthquakes, landslides, volcanic activities etc. in various scales, local, regional and global (Grezio et al., 2017). For the earthquake generated tsunamis, the method is generally adapted from seismic hazard assessment methods (González et al., 2009). Such kind of studies consider the events that are generated by co-seismic sea floor displacement, Seismic Probabilistic Hazard Analysis (SPTHA), but numerous tsunami simulations are required to consider all expected combination of seismic sources.
This problem can be solved by applying a simplified event tree approach and a two-stage filtering procedure to reduce the number of required source scenarios without decreasing the quality and accuracy of inundation maps (Lorito et al., 2015). The earthquake source itself is very uncertain and the investigation of this uncertainty can be done by building an event tree instead of using logic tree and hazard integrals (Selva et al., 2016). Logic tree approach can be applied to generation of tsunami hazard curves to decrease the uncertainties by including branches, which are the combination of tsunami sources, magnitude
distribution of characteristic tsunamigenic earthquakes, their recurrence interval, and tsunami height estimation procedure based on a numerical simulation (Annaka et al., 2007). For regional studies, hazard curves can be generated by empirical analysis using available tsunami run-up data. However, if such data is not available, MC simulations, a computational based method widely used in probabilistic seismic hazard analysis (PSHA), can be considered as a primary method to generate tsunami hazard curves (Geist and Parsons, 2006; Horspool et al., 2014). Submarine landslides, on the other hand, are the major
tsunami source for passive margins, which are the transition zone between oceanic and continental lithosphere that is not an active plate boundary,  and they have been included in PTHA methodologies (Geist and Lynett, 2014). Probabilistic studies are also applied to develop multi − hazard loss estimation methodology for coastal regions that are exposed to cascading shaking-tsunami hazards due to offshore mega-thrust subduction earthquakes (Goda and Risi, 2018).

In this study, characteristic earthquake model is used to estimate the earthquake recurrence on PIF. Paleoseismologic studies
(Ryall et al., 1966; Allen, 1968; Schwartz and Coppersmith, 1984) suggest that an individual fault tends to generate characteristic earthquakes having a very narrow range of magnitudes. These individual faults have a different frequency distribution than the log linear Gutenberg-Richter frequency-magnitude relationship (Aki, 1984; Schwartz and Coppersmith, 1984; Youngs and Coppersmith, 1985). According to Aki (1984), characteristic earthquake is generated as a result of constancy of barriers to rupture through repeated seismic cycles.

PIF is fully characteristic and a characteristic earthquake will rupture entire fault as a whole and release the entire energy. Therefore, while performing MC simulations, area of the fault and fault parameters (strike, dip and rake angles) are used as constants referring to the outcomes of EU 7th Frame Project MARSITE (Ozer Sozdinler et al., 2019). One of the work packages of this project aimed to define the geometry of the possible tsunamigenic faults in the Marmara Sea and 30 different earthquake scenarios with the different rupture combinations of 32 possible fault segments. Based on these 30 different earthquake
scenarios, tsunami numerical modelling is performed. The definition of fault segments depends on extensive review of the

literature (Alpar and Yaltırak, 2002; Altınok and Alpar, 2006; Armijo et al., 2005; Ergintav et al., 2014; Gasperini et al., 2011; Hebert et al., 2005; Hergert et al., 2011; Hergert and Heidbach, 2010; Imren et al., 2001; Kaneko, 2009; Le Pichon et al., 2001; Le Pichon et al., 2003; Le Pichon et al., 2014; Oglesby and Mai, 2012; Sengor et al.,2014; Tinti et al., 2006; Utkucu et al., 2009). As a result of this review, each fault segment is defined as a rectangular area with hypothetical uniform slip. According to the results of the project, the fault parameters of the PIF, are given in Table 1. The 3D Fault configuration given by Armijo et al., 2002, which explains fault segmentation in the region depending on morphology, geology and long-term displacement fields, also fits with the PIF parameters that are used in the project. These parameters are used as constants in this study while assessing probability of occurrence of each earthquake to allow entire fault rupture at different depths with different magnitudes.

| Fault Length (km) | Fault Width (km) | Strike | Dip | Rake |
|---|---|---|---|---|
| 33.5 | 14 | 119 | 80 | 210 |

**Table 1: The area and the focal mechanism of the PIF zone. These are the constant parameters during the MC simulation application.**

MC simulation technique is generally applied to generate earthquake catalogue of a given length of time. In this technique, a list of earthquakes can be generated using the frequency - magnitude relationship for each seismic source (Zolfaghari, 2015). Seismic zonation should be done by considering regions that have relatively homogeneous earthquake activity and faulting regimes (Sørensen et al., 2012). In this study, fault segment model proposed in Ozer Sozdinler et al., 2019 is used and PIF is the only segment for seismic source. After that, tsunami numerical modelling is performed for each event of this synthetic catalogue and tsunami hydrodynamic parameters, mainly maximum wave heights, inundation depth, current velocities, as well as tsunami inundation zones are estimated. Tsunami risk assessment will serve the best for the needs of societies when associate regional studies with the local ones (Sørensen et al., 2012).

MC simulation technique allows generating a list of earthquakes based on a frequency-magnitude relationship. This technique depends on a uniformly distributed source model and it provides equal chance to each earthquake source. As a result, synthetic earthquake catalogue will have uniformly random distributed earthquake sources (Zolfaghari, 2015).

Using MC simulation, a synthetic earthquake catalogue is generated by selecting earthquake magnitude and depth as uniformly distributed random numbers in a given interval and using area and directivity of the fault as a constant variable (Table 1). We performed MC simulations 100 times for having 100 different earthquake scenarios. The number of earthquakes in the catalog is selected as a reasonable number, which represents the number of iterations randomly done in MC simulations for having a synthetic earthquake scenario. As mentioned earlier, NAFZ generates an earthquake with the recurrence interval of about 250 years beneath the Marmara Sea. Therefore, selecting 100 earthquake scenarios would cover a time period of 100x250 years= 25,000 years which is considered as an adequate catalog duration in this study. However, because of having time dependent

probabilistic analyses, this catalog duration is not used for PTHA in this study.

Earthquake magnitude is one of the parameters randomly selected by the MC technique. Based on a characteristic earthquake model, individual faults tend to rupture entire fault when a large earthquake occurred. This model assumes that characteristic earthquake releases all of the seismic energy during the fault rupture and the magnitude of the earthquake depends on the dimension of fault (Abrahamson and Bommer, 2005).

As mentioned previously, only the PIF is considered as earthquake source with approximately 34 km in length and 14 km in width (Ozer Sozdinler et al., 2019; Karabulut et al., 2002). This fault zone is assumed that it has potential to generate a characteristic earthquake and rupture the entire fault. According to Wells and Coppersmith (1994) scaling relation between fault area and magnitude (Eq. 1), this fault can generate a characteristic earthquake with the magnitude varying between $M_w$ 6.5-7.1.

$$M_w = a + b * \log (L * W) \tag{1}$$

In this equation, a and b are coefficient, which are 4.33 and 0.9 respectively, L is fault length and W is the fault width.

Displacement on the fault surface calculations are done, for each randomly selected magnitude, using the formulation of Aki (1966),

$$D = \frac{M_o}{\mu A} = \frac{10^{(M_w + 6.07)*1.5}}{\mu A} \tag{2}$$

where D is displacement on the fault surface, $M_w$ is moment magnitude, μ is the shear modulus (μ=30 GPa), and A is the fault area.

Seismogenic thickness and the location of the earthquake is another important parameter required for earthquake and tsunami source. At first, the PIF zone is accepted as fully characteristic and an earthquake should rupture the entire fault area. Therefore, it is assumed that if the rupture starts at the center of the fault and continues in both directions, the fault will rupture entirely. For this reason, the locations of the earthquakes are accepted as the midpoint of PIF zone for each earthquake scenario (Ozer Sozdinler et al., 2019).

For the seismogenic thickness, the seismic activity of the northern segment of NAFZ starts at the depth of 5 km (Karabulut et al., 2003). The bottom of the seismogenic thickness can be determined based on the after-shock activity of the 17 August 1999 Izmit Earthquake. The earthquakes on the northern scarp of the Çınarcık basin are observed between the depths of 5 and 14 km. The mechanism of events between the depth of 5 and 10 km shows the behavior of normal faulting. On the other hand, strike-slip mechanism dominates the depths below 10 km to 14 km. As a result, seismic activity can be observed between the depths of 5-14 km and fault plane solutions show normal and strike-slip mechanisms in this area (Karabulut et al., 2002). Therefore, depth of events vary between 5 to 14 km in MC simulations.

In time - independent earthquake occurrence models, probability of an event occurrence follows a Poisson distribution in a given certain period of time. Therefore, the result of this model does not vary in time. However, probability of an earthquake occurrence is based on the time that has passed since the occurrence of last event and it follows a Brownian passage time (BPT), log-normal or another probability distribution (Matthews et al., 2002; Ellsworth et al., 1999; Davis et al., 1989; Rikitake

1974). In this model, in addition to the recurrence time of earthquake, variability of the frequency of events and the elapsed time from the last characteristic event are the additional required information and the longer elapsed time causes to increase of probability of an event occurrence (Cramer et al., 2000; Petersen et al., 2007).

Calculation of probability in multi – segment ruptures and more complicated models includes Gutenberg Richter magnitude – frequency relationship (Gutenberg and Richter, 1944). The application of time – dependent models based on characteristic earthquake model, which assumes all large events occurring along a particular fault segment would have similar magnitudes, rupture area and average displacements (Schwartz and Coppersmith, 1984). Therefore, this model is suitable for calculating the probability of occurrence of an earthquake on a single fault.

It should be noted that, in this study, PIF is considered as the only source for the earthquake and tsunami. Time dependent probabilistic model is followed for the probability calculations; because, this probabilistic model allows to consider only one fault instead of using multi – segment rupture scenarios through characteristic earthquake model.

In the time-dependent approach, Brownian passage time (BPT) probability model is used to obtain the recurrence time probability of the earthquake in the fault segment. This model does not show significant difference with the log – normal distribution except for consideration of very long elapsed times from the last characteristic event (Petersen et al., 2007). A characteristic event occur when the load-state process reaches to the failure threshold; an earthquake releases all energy loaded on the fault and then starts the new failure cycle. The time interval between consecutive earthquakes shows a Brownian passage time distribution and that can be useful to forecast long term seismic events by generating a time – dependent model (Matthews et al., 2002). The Working Group on California Earthquake Probabilities (1999) and the Earthquake Research Committee (2001) have already implemented this time – dependent approach to the San Francisco Bay and Japan, respectively, for the prediction of long-term events (Petersen et al., 2007). This model depends on the time period passed since the last characteristic event and recurrence time of the earthquake. The probability density function for BPT model (Matthews et al., 2002) is given by,

$$f(t, T_r, \propto) = (\frac{T_r}{2\pi\alpha^2 t^3})^{1/2} \exp (\frac{(t-T_r)^{1/2}}{2T_r\alpha^2 t}) \tag{3}$$

where t is the elapsed time from the last characteristic event and $\alpha$ is the aperiodicity (also known as the coefficient of variation). Aperiodicity defines the regularity of the expected characteristic earthquakes on the fault and varies between the 0.3 and 0.7. This parameter, which is known as the parameter defining how much an expected characteristic earthquake occurs regularly or irregularly on any fault segment (Murru et al., 2016), was taken as 0.5 in this study (Parsons, 2004). The mean recurrence interval of earthquakes ($T_r$) can be defined as the ratio between the mean moment of repeating earthquakes (seismic moment) and the long-term moment accumulation rate on the fault (moment rate) (Ren and Zhang, 2013). Seismic moment can be obtained using the formulation of Kanamori (2004) and the moment rate of the fault is calculated from fault area and long-term slip rate of the fault (WGCEP 2003).

$$T_r = \frac{M_o}{\dot{M}_o} = \frac{10^{(M_w+6.07)*1.5}}{\mu V A} \tag{4}$$

In this equation, $M_w$ is moment magnitude, $\mu$ is the shear modulus, V is long-term slip rate in mm/yr and A is the fault area.

The moment magnitude value in Eq. (4) was selected randomly using MC simulations. Thus, seismic moment ($M_o$) and the mean recurrence time ($T_r$) were calculated for each earthquake scenario. Long term slip rate is also selected as 17 mm/yr for

this equation (Ergintav et al., 2014).

Probability of the earthquake occurrence on the fault is calculated based on the probability density function approach. The probability of occurrence of an event in the next ΔT years, given that it has not occurred in the last t years is given by (Erdik et al., 2004),

$$P(t, \Delta T) = \frac{\int_t^{t+\Delta T} f(t)dt}{\int_t^{t+\infty} f(t)dt} \tag{5}$$

In this case, probability of a characteristic earthquake was calculated using ΔT as 50 and 100 years.

## 3 Tsunami Numerical Modelling

Tsunami simulations are performed for each earthquake in synthetic catalogue using tsunami numerical model NAMI DANCE (NAMI DANCE, 2011). The code is the user-friendly version of TUNAMI-N2 (Imamura et al., 2001) developed in C++ language, which computes all fundamental parameters of tsunami motion in shallow water and in the inundation zone. It uses

explicit numerical solution of shallow water wave equations with finite-difference technique and allows for better understanding of the effect of the tsunami waves (Shuto et al., 1990; Imamura, 1989). NAMI DANCE can solve both Linear and Nonlinear Shallow Water (NSW) Equations with selected coordinate system (Cartesian or spherical) and calculates the tsunami motion. LSW equations are preferable in deep water because of reasonable computer time and memory and calculates the results in acceptable error limit (Insel, 2009). NAMI DANCE is validated and verified using NOAA standards and criteria

for tsunami currents and inundation (Synolakis et al., 2007; Synolakis et al., 2008). The numerical solutions of NAMI DANCE are also tested, validated and verified against analytical solutions, laboratory measurements and field observations (NTHMP, 2015; Lynett et al., 2017; Velioglu, 2009)

NAMI DANCE calculates tsunami generation using Okada (1985) equations. In this study, water surface distribution of tsunami source (initial wave amplitude) are calculated with this method for 100 earthquakes of the synthetic earthquake

catalogue prepared by MC simulations. As an example, Figure 5 shows the initial water surface calculated due to one of 100 tsunami sources generated by MC simulations (Fig 5).

Before starting tsunami simulations, the necessary inputs should be prepared precisely in order to obtain reliable results. Bathymetry - topography data is one of the most important input in NAMI DANCE that significantly effects the reliability of results especially in shallow water zone due to the nature of NSW Equations. NAMI DANCE can make nested analyses under

the condition that grid size of study domains have a certain 1:3 ratio between each other. Therefore, we generated four nested domains having the coarsest grid size as 81 m and the finest grid size as 3 m with 1:3 ratio in GIS environment. Bathymetric data for the biggest domain is the combination of 30 arc second resolution General Bathymetric Chart of the Oceans (GEBCO) and data produced by navigational charts in shallow zones. Topographic data, on the other hand, contains the high resolution, which is obtained from the Department of Housing and Urban Development of Istanbul Metropolitan Municipality, Digital

Elevation Model (DEM) and vector data with resolution 5m and 1m, respectively. The bathymetry - topography data in the smaller domains is the downscaled version of the 81m grid sized bathymetry-topography data, however high resolution digitized coastline, and sea and land structures are also included to the data to generate 3m grid sized smallest domain (Fig 4). Synthetic gauge point file is another required input of the NAMI DANCE. In addition to the calculation of principal tsunami hydrodynamic parameters, program can also calculate the change of water level, current velocity and flow depth over time in

every gauge point. Therefore, various gauge points are selected along the coast of nested domains, near shore and offshore and close to some critical structures on land.

    During the inundation of tsunami waves, current velocity is an important tsunami parameter in land and sea, especially in ports and bays. Strong current velocities may cause dragging offshore or landing of sea vessels inland. This parameter as well as tsunami wave amplitude, inundation depth and Froude number can be calculated by NAMI DANCE. However, in this study,

the results are represented based on only the probability of exceedance of threshold values for water surface elevation and inundation depth.

    NAMI DANCE can make nested analyses under the condition that grid size of study domains have a certain 1:3 ratio between each other. Therefore, we generated four nested domains having the coarsest grid size as 81 m and the finest grid size as 3 m with 1:3 ratio in GIS environment. Coarser data includes multi-beam bathymetric measurements and 900m grid sized GEBCO

data in the sea and 30m grid sized ASTER data on land. Coastline, and sea and land structures are also digitized in GIS environment and included in 3m grid sized high resolution bathymetry - topography data in the smallest domain (Fig 4).

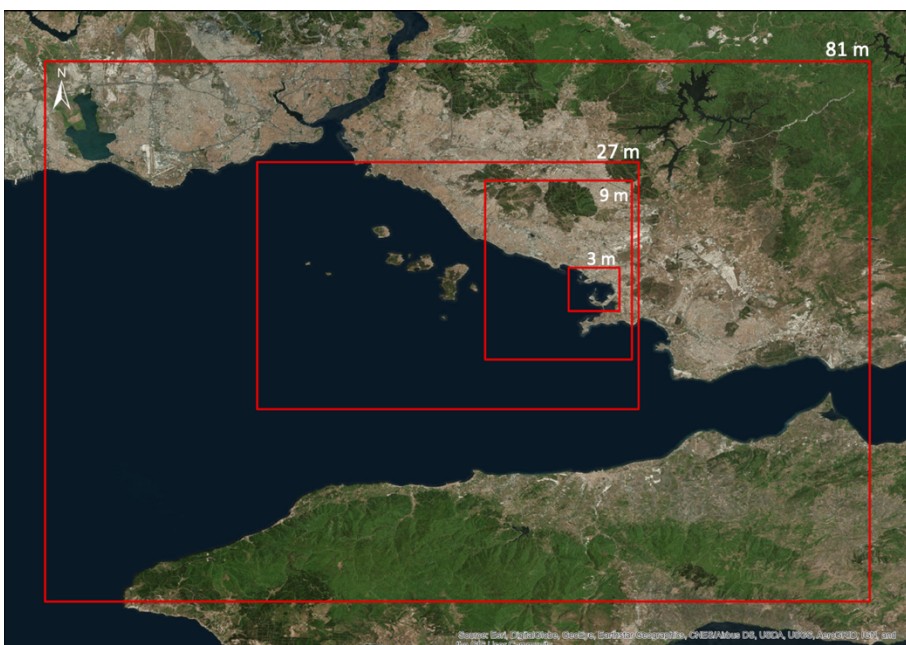

**Figure 4: Nested domains for tsunami numerical modelling. Red rectangles show the limits of these domains. Grid size of these domains have a certain 1:3 ratio between each other (created using ArcMap Version 10.5).**

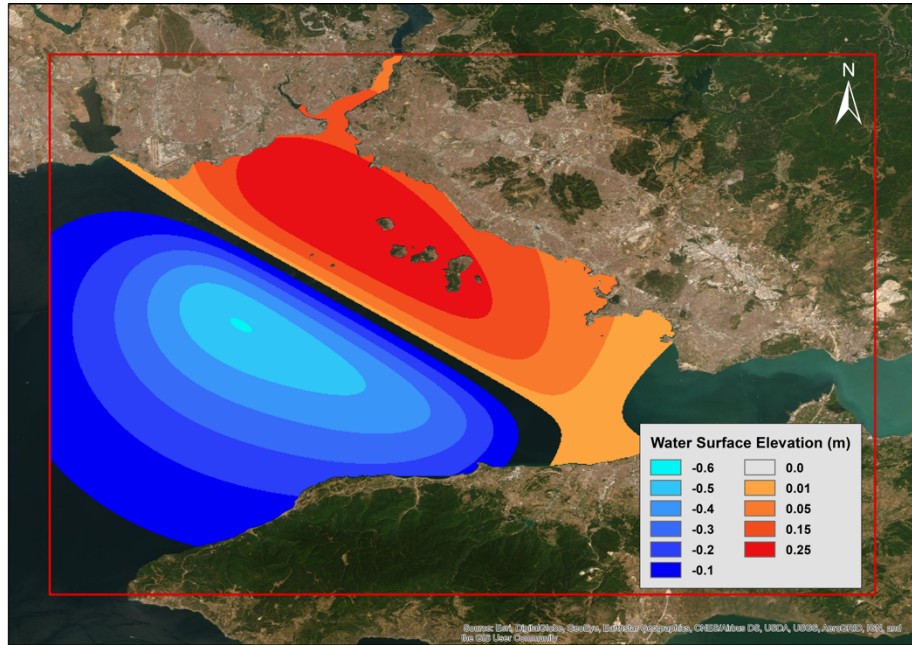

**Figure 5: Initial water surface distribution of one of the 100 tsunami sources. Red frame indicates the boundary of largest nested domain with 81m grid size (plotted using ArcMap Version 10.5).**

**4 Results and Discussion**

In this study, tsunami hydrodynamic parameters are calculated in both coarsest domain (whole Marmara Sea) and finest domain (Tuzla region). The main parameters focused in this study are the tsunami wave heights and inundation depths and the results are shown in the terms of probability of exceedance of threshold wave height and inundation depth values within the next 50 and 100 years. The situation for the next 500 years is not considered because the return period of the fault rupture is about 250 years, which means this fault generates at least one earthquake within the next 500 years. In other words, probability of exceedance for the next 500 years will be about 99%.

We present the results of the PTHA for Tuzla test site in terms of three different visualization categories for the next 50 and 100 years. First, distribution of probability of occurrence of the tsunami hydrodynamic parameters, which are minimum and maximum water surface elevation and inundation depth, are shown. Second, tsunami inundation maps that show probability of exceedance of 0.3 m inundation depth for different time periods are generated for Tuzla region in order to observe flooded areas and their probabilities clearly. And finally, the probability map of exceedance of 0.3 m wave heights at synthetic gauge points are represented as bar charts.

**4.1 Probability of Exceedance for Entire Synthetic Earthquake Catalogue**

The graphics are generated to demonstrate the probabilities of occurrences corresponding to the minimum and maximum water surface elevations and inundation depth calculated from tsunami sources of each earthquake in synthetic earthquake catalogue. It should be noted that in case of having same magnitude of earthquakes in two different earthquake scenarios of the catalogue,

the probability of occurrences of these scenarios would be the same. However, since they would have different focal depths, the tsunami initial wave height calculated by Okada (1985) will be different, which results in the calculation of different hydrodynamic parameters. As a result, the graphs show different maximum water surface elevations having the same probability of occurrences.

In Figure 5, graphics of probabilities of occurrences according to maximum and minimum water surface elevation (maximum water withdraw) and inundation depth for next 50 years are represented, respectively. According to these graphs, tsunami wave heights up to 1 m and withdrawal of the waves around 1 m have approximately 65%±15 probability of occurrence. Tuzla region includes various shipyards, ports and other important facilities. Therefore, probability of the withdrawal of the water is as important as maximum water surface elevation. 1 m height of wave withdrawal may cause the ships to be stranded at the

ports and results in extreme financial losses as observed in the 20[th] July 2017 Bodrum-Kos earthquake and tsunami (Yalçıner et al., 2017). The probability for having 1 m inundation depth, on the other hand, can be predicted as about 60%±10. Residual of probability with respect to the fitted curve for each data point is demonstrated right after the percentage of probability with ± sign.

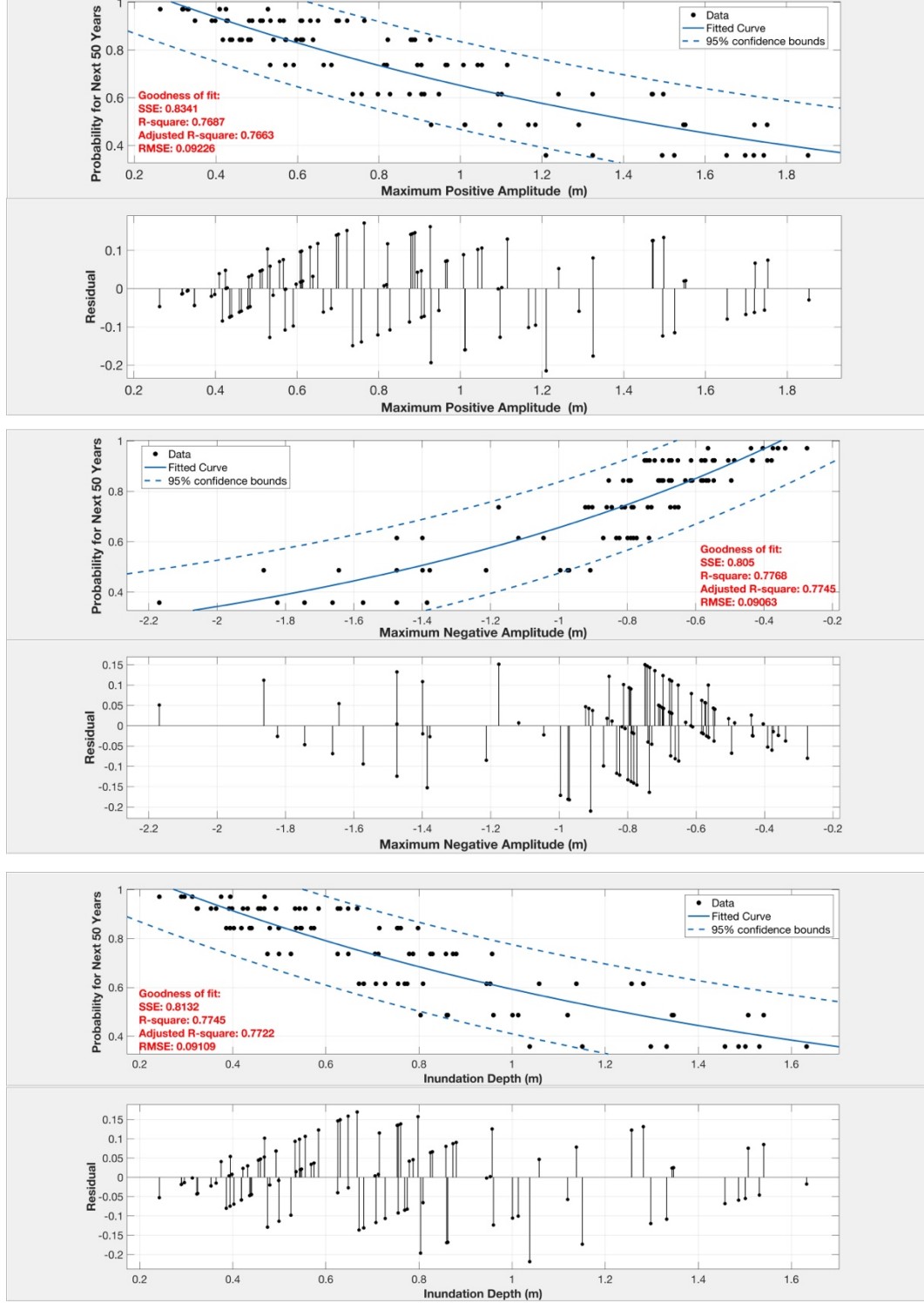


**Figure 5: Probabilities of Exceedance Corresponding to Maximum Water Surface Elevation, Minimum Water Surface**

**Elevation and Inundation Depth for the next 50 years. Black dots represent the probability of exceedance of tsunami hydrodynamic parameter for each event in the catalog. Blue line is the best fit curve to the data and dashed blue line is**
**the 95% confidence boundary of fitted curve. Residual of the fit is represented for each probability curve.**

The situation for next 100-years (Fig 6) obviously shows that probability of occurrences would increase with time. The probability of exceedance of 1 m water surface elevation and 1 m wave withdrawal reaches up to 85%±10. Probability of exceedance of inundation depth also changes significantly. The probability of exceedance of 1 m inundation depth is found
around 80%±10.

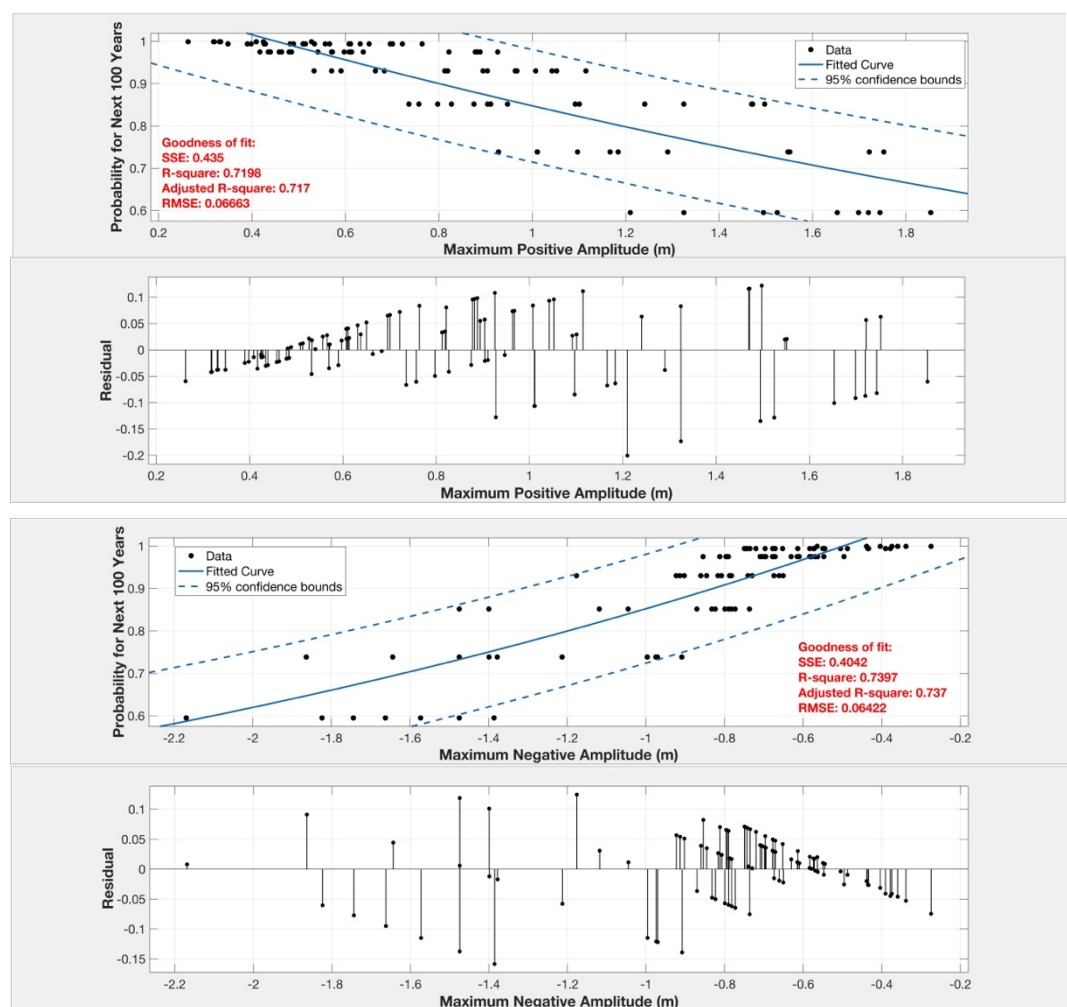

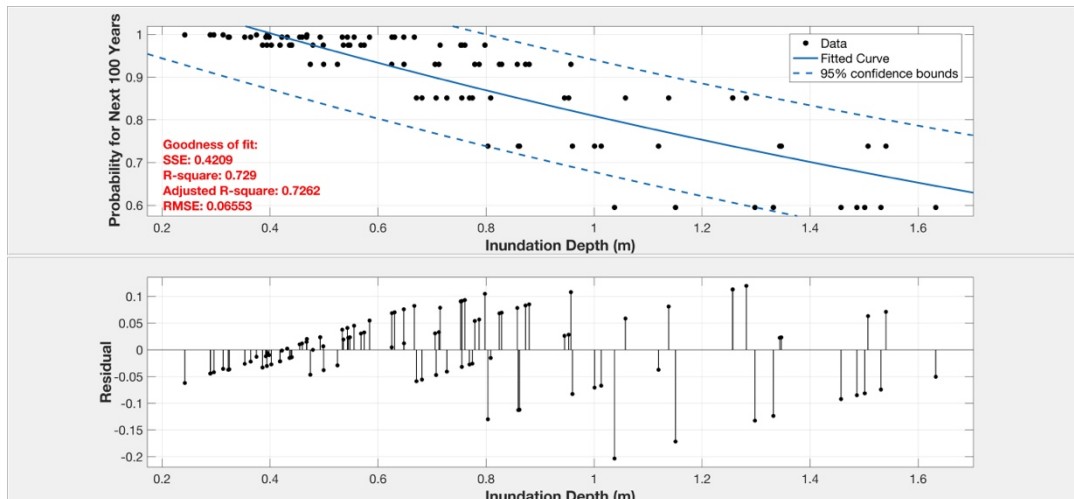

**Figure 6: Probabilities of Exceedance Corresponding to Maximum Water Surface Elevation, Minimum Water Surface Elevation and Inundation Depth for the next 100 years. Black dots represent the probability of exceedance of tsunami hydrodynamic parameter for each event in the catalog. Blue line is the best fit curve to the data and dashed blue line is the 95% confidence boundary of the fitted curve. Residual of the fit is represented for each probability curve.**

Considering the results of the whole simulation, the worst case earthquake scenario generated tsunami waves with maximum water surface elevation equal to 1.8 m, minimum water surface elevation (maximum withdraw) equal to -2.1 m and inundation depth equal to 1.6 m. The probability of occurrence of this event is 35% for next 50 years and 60% for next 100 years.

**4.2 Probabilistic Tsunami Inundation Maps for Tuzla Test Site**

Inundation maps of Tuzla domain are also prepared for the next 50 and 100-years in GIS environment. Even if inundation depth is in the order of few centimeters, it can lead to dragging of people in coastal regions due to the high current velocities of the waves (Jonkman and Penning-Rowsell, 2008). Therefore, these inundation maps have a great significance to understand the flooded areas in study domain and the amount of water penetrated inland.

Generation of inundation maps are based on the probability of exceedance of 0.3 m inundation depth. There are several studies in literature proving both experimentally and numerically that tsunami waves with the order of 0.3 m height have a potential to collapse a human body (Jonkman and Penning-Rowsell, 2008; Takagi et al., 2016). For this reason, only the earthquake scenarios that generated inundation depths larger than or equal to 0.3 m threshold value are considered.

- Inundation depth files, which is one of the outputs of the NAMI DANCE, are used for the calculation.
- The inundation depth values at each grid node are replaced with the probability of occurrence of the respective earthquake scenario. We repeated this procedure for all earthquake scenarios, which has inundation depths larger than or equal to 0.3 m threshold.

The mean (average) probability of occurrence is calculated at each grid node. Thus, the spatial distribution of probability of

exceedance of 0.3 m inundation depth in inundation zone is obtained for a specific time interval (Fig 7).

Figure 7 shows the inundation maps of Tuzla shipyard for the next 50 and 100-years. Most of the area in Tuzla shipyard region have probability of exceedance between 10% and 20% for both of the next 50 and 100 years. However, some places in the northern and southern part of the area and inside the bay show larger than 75% probability of inundation within the next 100 years. Maximum inundation distance is observed as around 60 m in the test site.

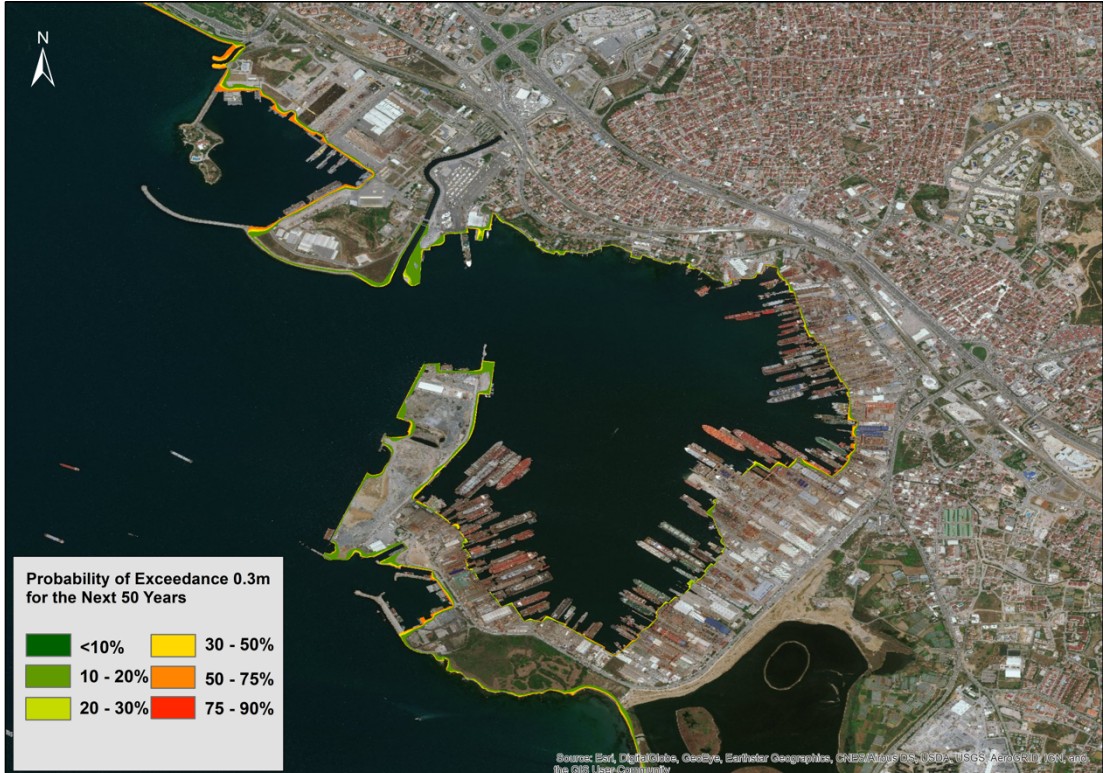

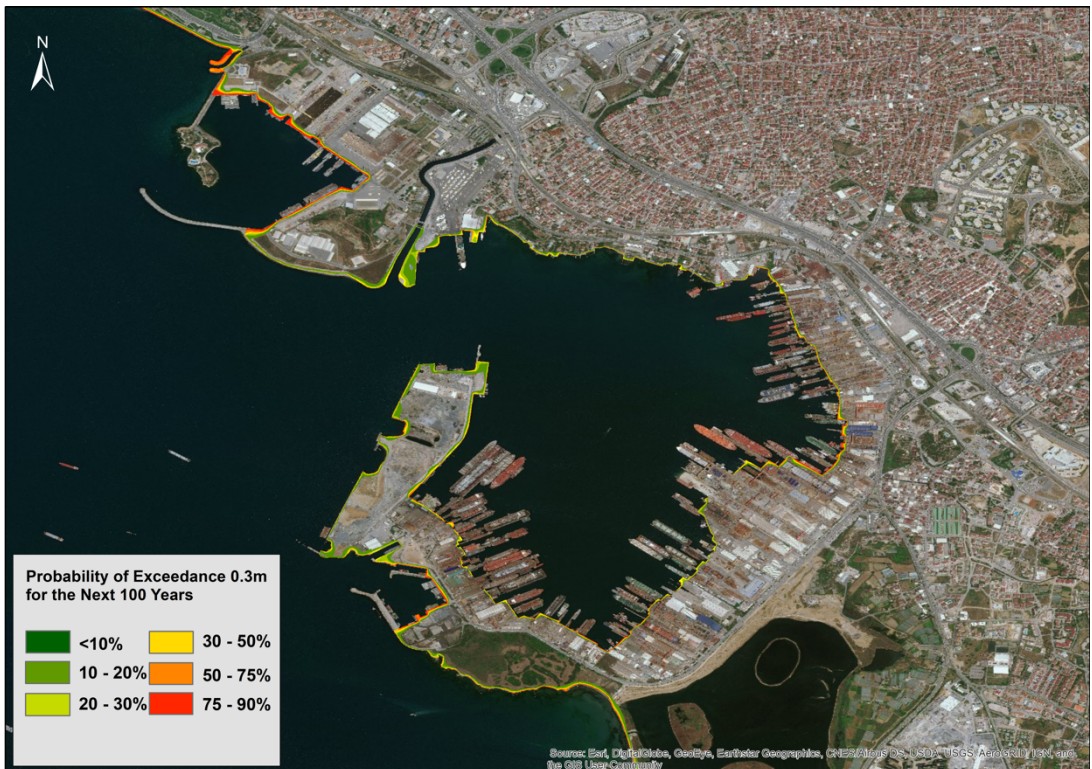

**Figure 7: Probabilistic Tsunami Inundation Maps for Tuzla Study Domain representing the Probability of Exceedance of 0.3 m Inundation Depth within the Next 50 and 100 Years. Change of colors from green to red represents the increasing probability of exceedance (created using ArcMap Version 10.5).**

In Figure 8, probabilistic inundation maps of one of the most important facilities in the study region are represented for the next 50 and 100 years. The area has high potential to be exposed to tsunami waves with probability larger than 50% for the next 50 years. In 100 years, this probability increases and varies between 75% and 90%. No significant inundation zone is observed along the coast of the seawall and the peninsula. This may be due to the high ground elevation of these zones. Tsunami waves are inundated up to 45 m inside the small bay. This inundation distance could cause severe damage to shipyard and other constructions if corresponding current velocities are also significant.

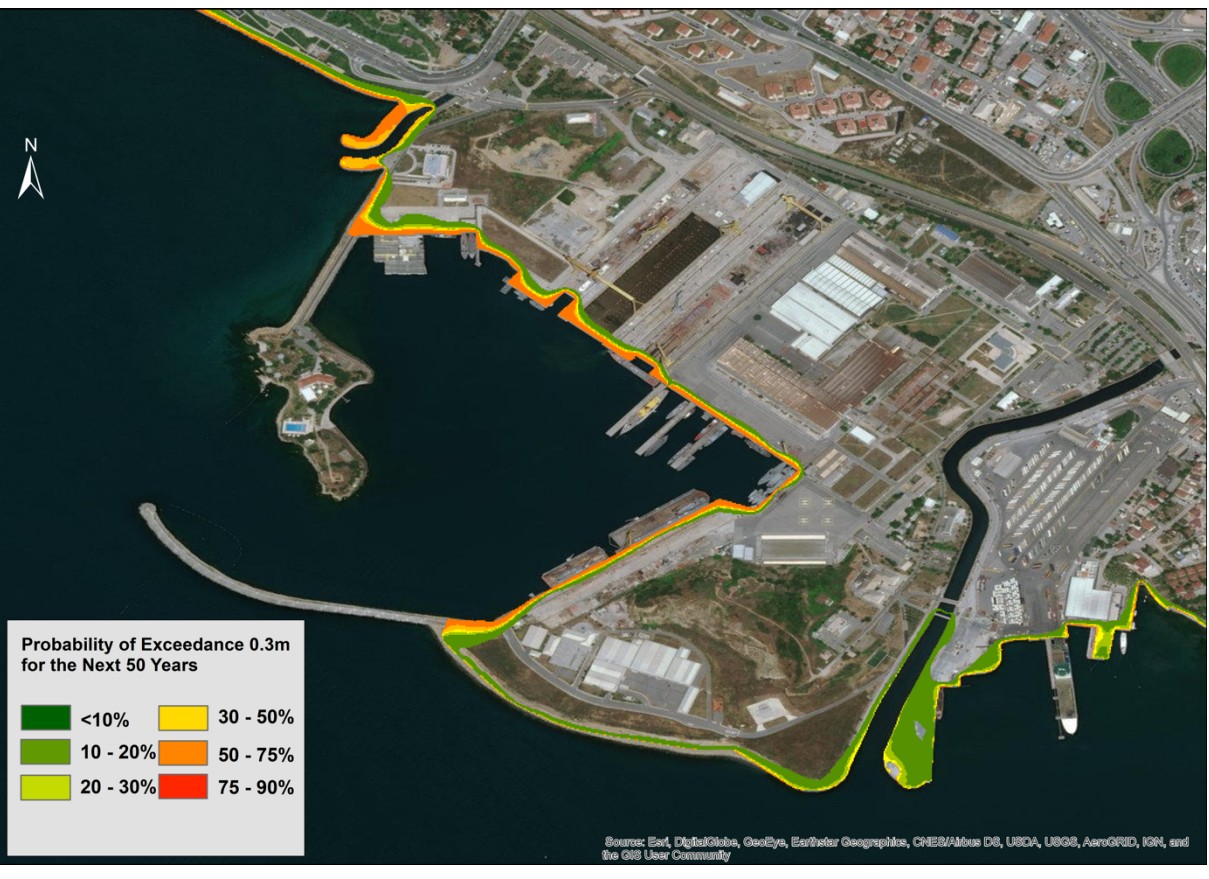

Probability of Exceedance 0.3m
for the Next 50 Years

<10%    30 - 50%
10 - 20%    50 - 75%
20 - 30%    75 - 90%

Source: Esri, DigitalGlobe, GeoEye, Earthstar Geographics, CNES/Airbus DS, USDA, USGS, AeroGRID, IGN, and the GIS User Community

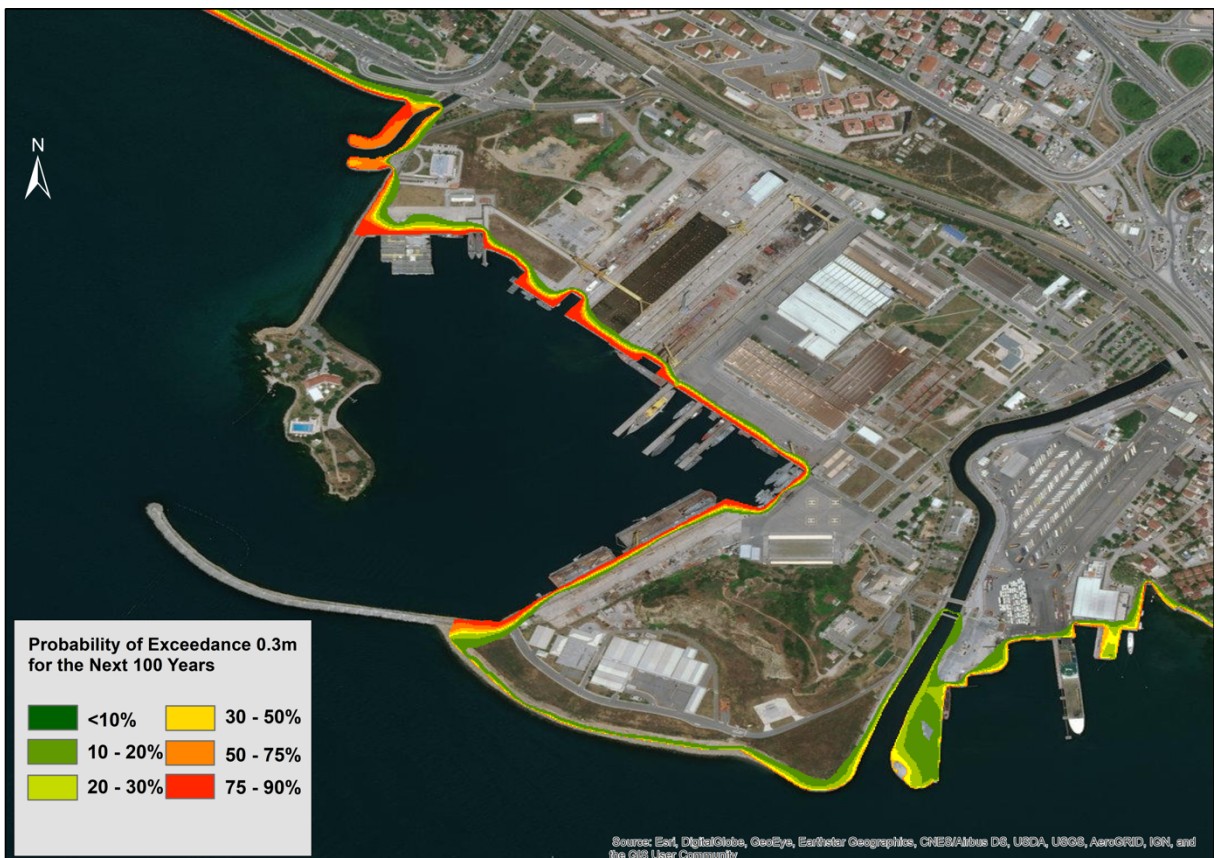

**Figure 8: Probabilistic Tsunami Inundation Maps of Northern Part of Tuzla Study Domain representing the Probability of Exceedance of 0.3 m Inundation Depth for the Next 50 and 100 Years. Change of colors from green to red represents the increasing probability of exceedance (created using ArcMap Version 10.5).**

In the next figure (9), the southern part of the Tuzla shipyard is seen according to probabilities of inundation for the next 50 and 100 years. Very limited area in the coastal zone is inundated with the probability between 30% and 50% within the next 50 years. The probability reduces up to 10% at some inner locations from the coastline. For 100-year recurrence time, the situation is almost the same. Only minor parts of the region at the south approaches up to 75% - 90% probability of exceedance of 0.3 m inundation depth threshold. The maximum inundation distance is calculated about 60 m. The inundated region does not include any important facility or structure and the effect of the tsunami will be minimal. The inundation distance decreases to 10 m at the other parts of the region.

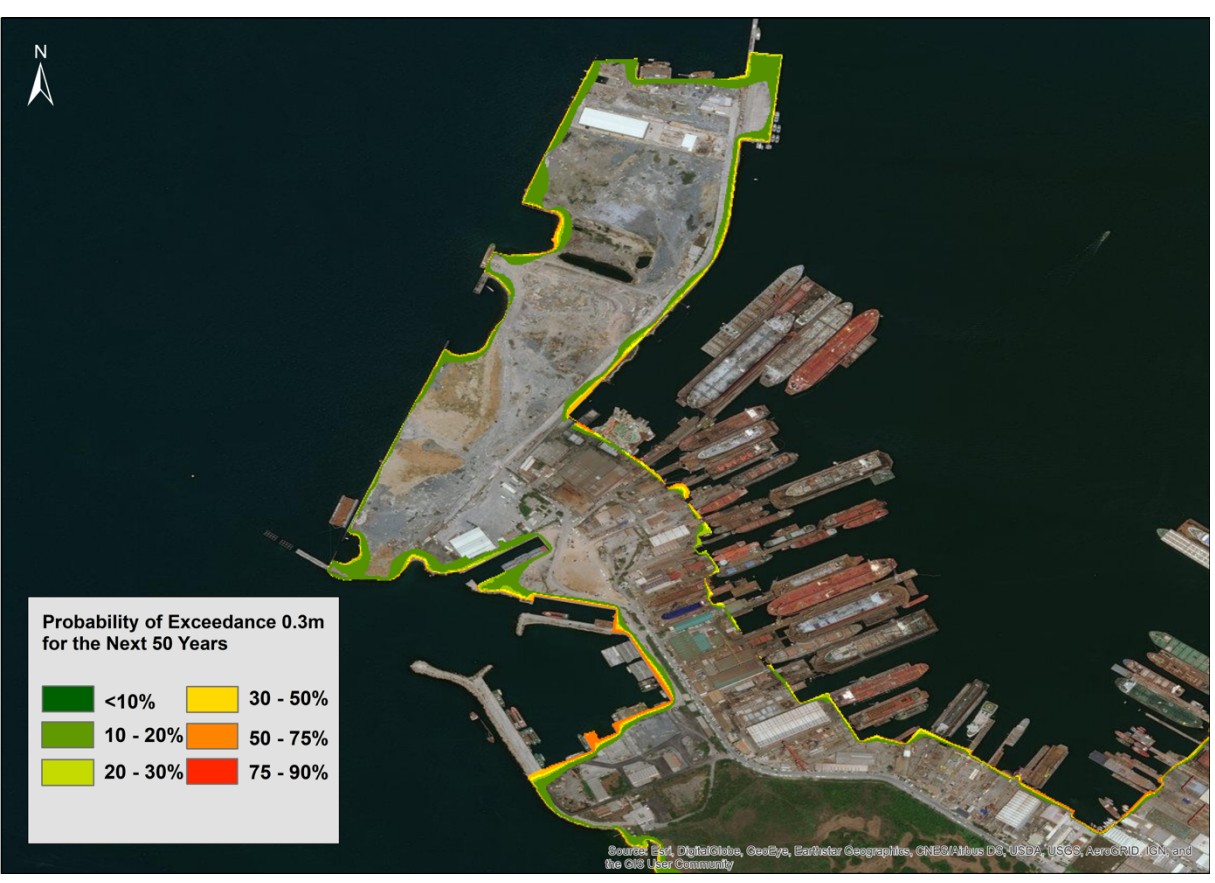

Probability of Exceedance 0.3m
for the Next 50 Years

- <10%
- 10 - 20%
- 20 - 30%
- 30 - 50%
- 50 - 75%
- 75 - 90%

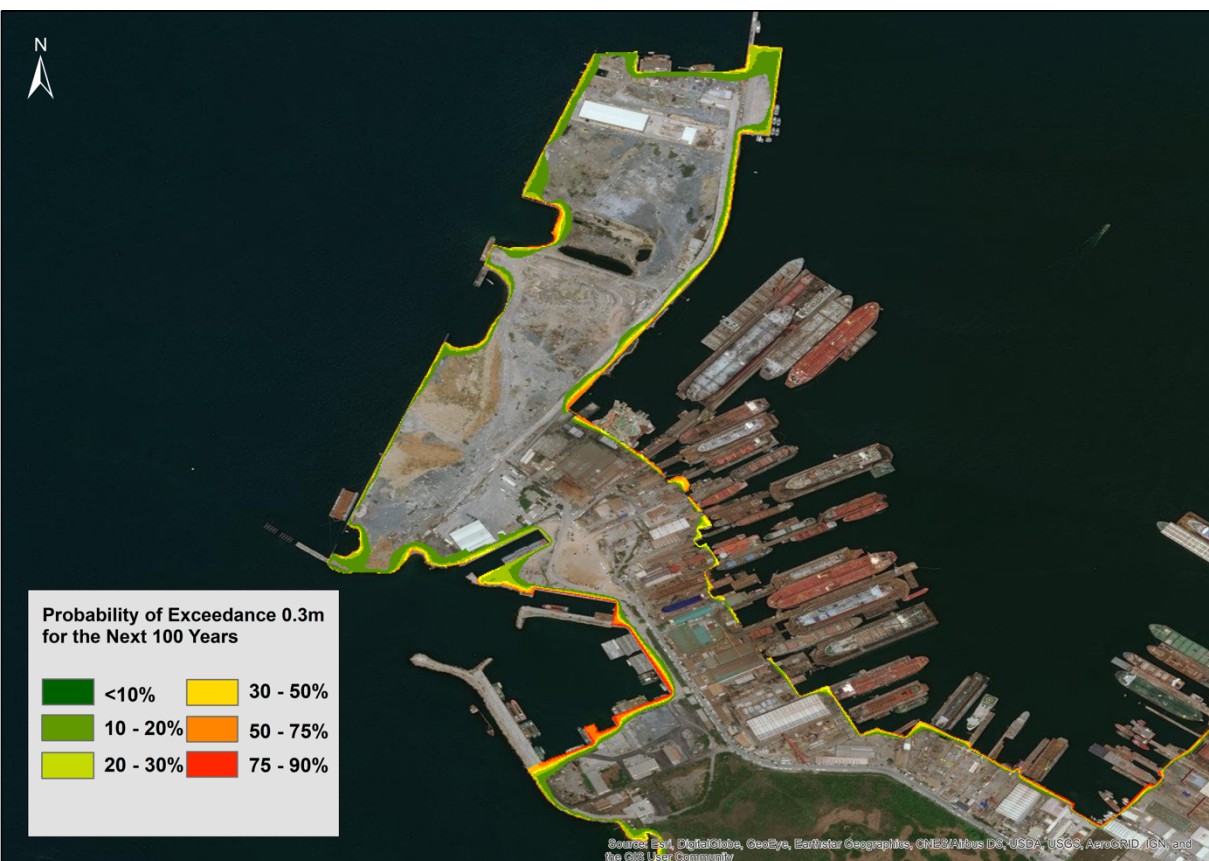


**Figure 9: Probabilistic Tsunami Inundation Maps for the Southern Part of Tuzla Study Domain representing the Probability of Exceedance of 0.3 m Inundation Depth for the Next 50 and 100 Years. Change of colors from green to red represents the increasing probability of exceedance (created using ArcMap Version 10.5).**

The region indicated in Fig 10 is located inside the bay and includes a large part of the shipyard area. This area includes lots

of large and small piers and ship construction facilities. The situation is more or less the same with the previous region (Fig 8). Probability of having larger than 0.3 m inundation depth changes between 30% and 50% within the next 50 years, while only a few places show 75% - 90% probability for the next 100 years along the coast. Moreover, maximum inundation distance is calculated as 25 m for this zone. Even if the probability of inundation is low, these zones should be taken into consideration before constructing a new structure.


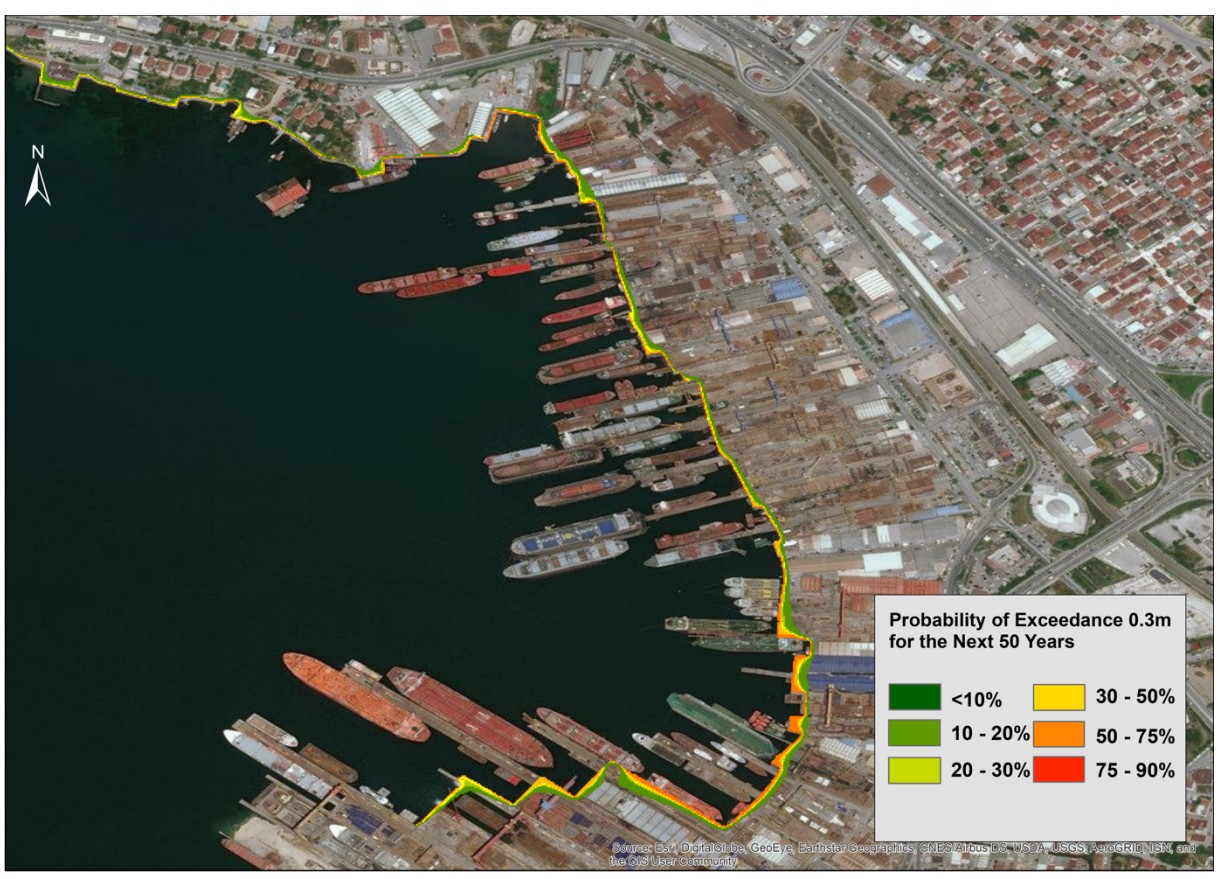

Probability of Exceedance 0.3m
for the Next 50 Years

<10%    30 - 50%
10 - 20%    50 - 75%
20 - 30%    75 - 90%

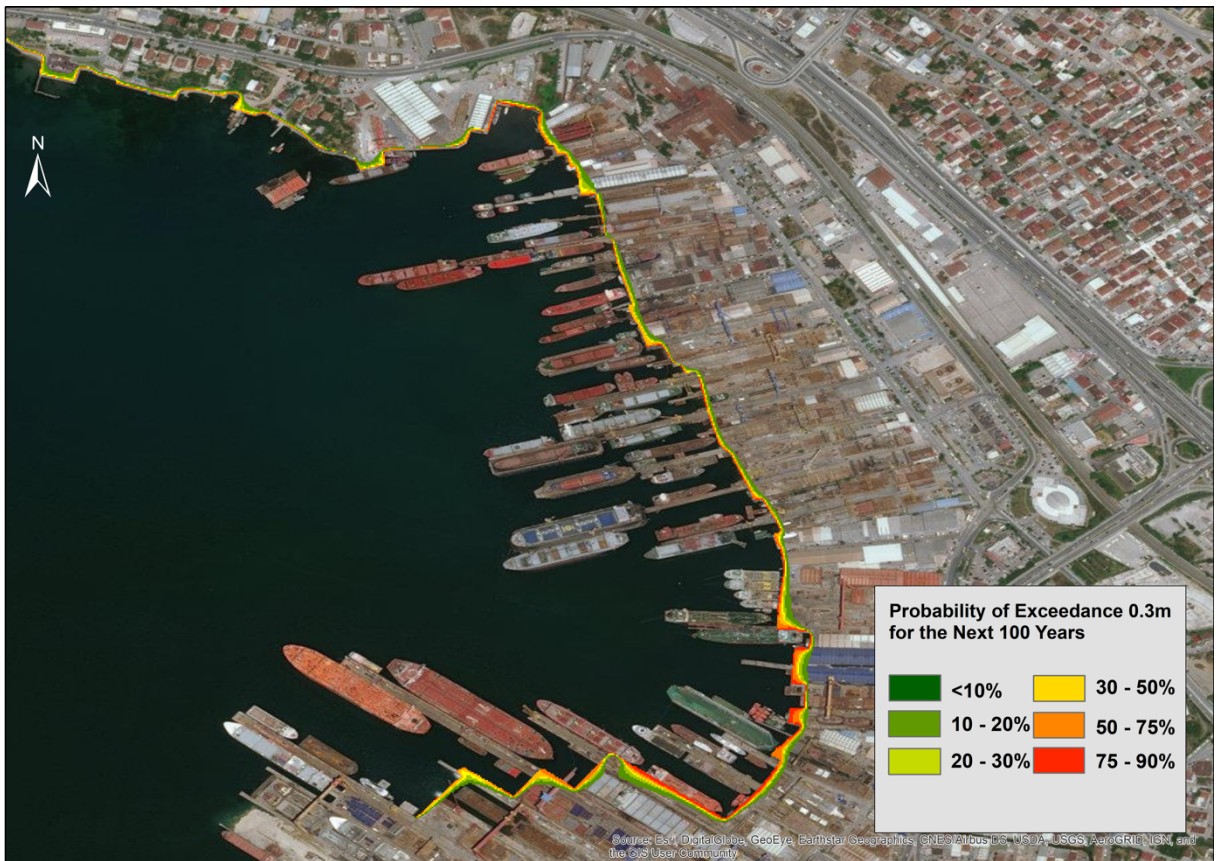

**Figure 10: Probabilistic Tsunami Inundation Maps of Shipyard Area in Tuzla Study Domain representing the Probability of Exceedance of 0.3 m Inundation Depth for the Next 50 and 100 Years. Change of colors from green to red represents the increasing probability of exceedance (created using ArcMap Version 10.5).**

### 4.3 Synthetic Gauges

Finally, the probability of exceedance of 0.3 m wave heights at synthetic gauge points are presented by bar charts to consider the near shore effect of tsunami waves along the western coast of Istanbul. Because of the closeness to the fault zone, the southeast coasts of the city are under the threat of the significant tsunami damage. Similar with the method applied during the preparation of probabilistic inundation maps, the earthquake scenarios with wave heights at synthetic gauge points larger than or equal to 0.3 m are selected and replaced with the probability of each scenario according to wave heights and after that the average probabilities at each synthetic gauge point are obtained accordingly.

Figure 11 demonstrates the probability of exceedance of 0.3m wave height at synthetic gauge points, which are about 350 m distant from each other, along the western coast of the Istanbul within the next 50 and 100 years. The probability is increasing while color scale chances from green to purple. According to this figure, minimum probability of exceedance is shown as 75% at some points. Except for a few of 228 synthetic gauge points, all points have larger than 90% probability of exceedance of

0.3 m wave height within the next 50-years time scale.

This condition is very serious since there are so many residential areas and important spots such as ports and recreational facilities in this region. The minimum probability of occurrence, which can generate tsunami waves with at least 0.3 m wave heights, reaches up to 90% for the next 100-year time period. However, 95% probability of exceedance of 0.3 m wave height dominates the region for this time scale.

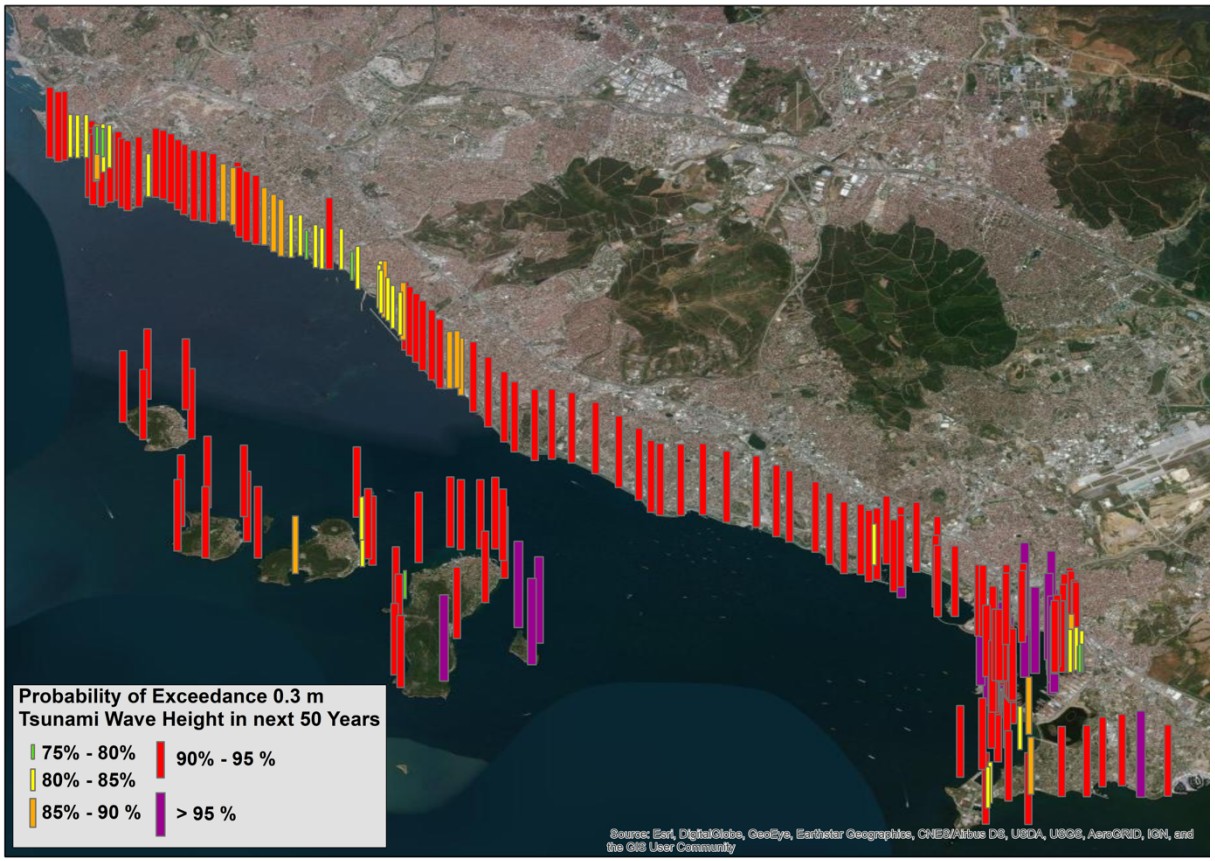

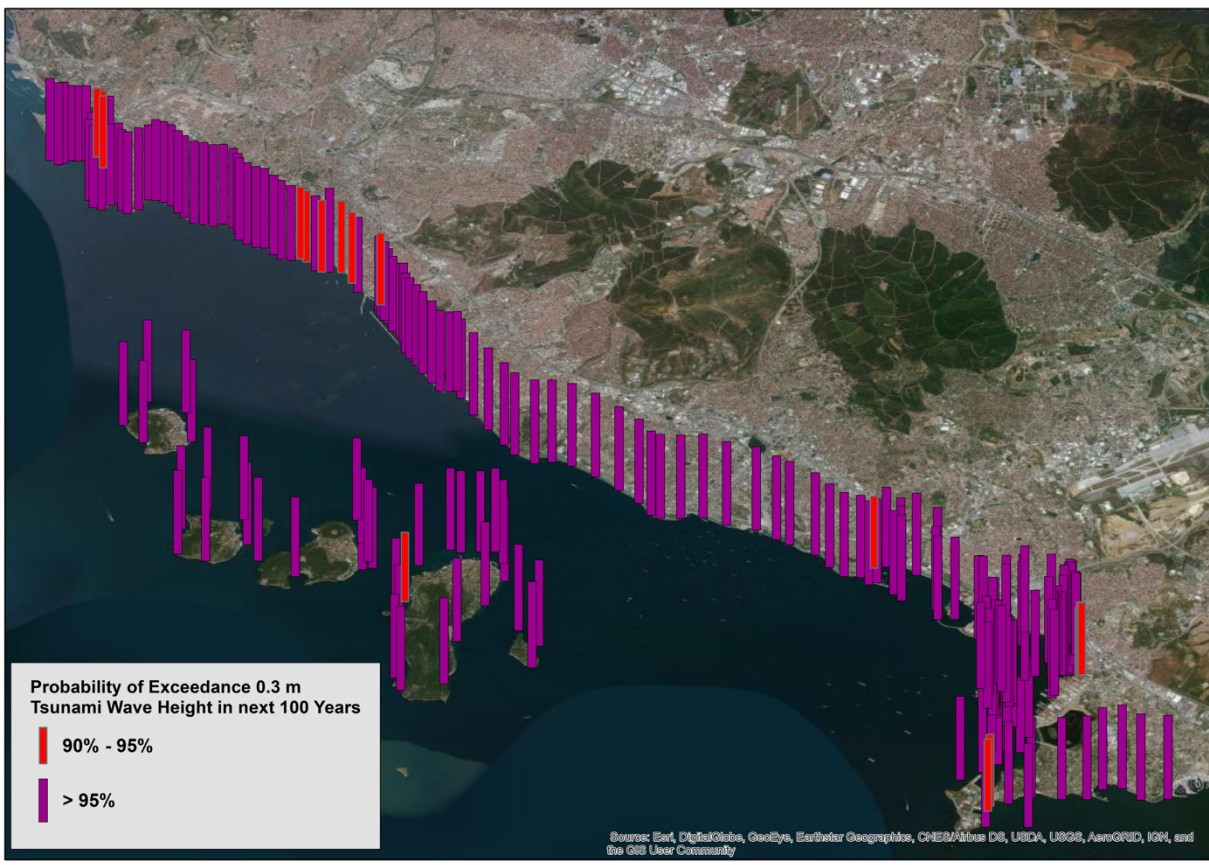


**Figure 11: Probability of Exceedance of 0.3 m Tsunami Wave Height within the next 50 and 100 years at Synthetic Gauge Points (yellow rectangles show the Tuzla study domain, change of colors from purple to green on the bars represents the decreasing probability of exceedance (created using ArcMap Version 10.5).**

**4.4 Uncertainties**

PTHA studies include some uncertainties because of the nature of rare occurrence the large events. Quantification of these uncertainties generally includes mixture of empirical analyses and subjective judgment.

Uncertainties of PTHA can be divided into two: as aleatory and epistemic variability. Aleatoric uncertainty is the natural randomness of the physical process. Including more data in the analyses does not contribute to reduce the aleatoric uncertainty.
However, knowledge about the modelling process may decrease this unpredictability. The occurrence time of the earthquake is one of the most fundamental aleatory variables in PTHA. This parameter is generally assumed a time-independent variable. However, in this study we used time-dependent probability model which reduces the uncertainty on this parameter. Mechanism of the source is considered as another aleatory variable for PTHA studies. The great number of the earthquakes all round the world occur at well-defined plate boundaries. However, some unidentified low active intraplate faults exist, which are recently
contained in PTHA studies (Selva et al., 2016). Moreover, the fault volume, which is used in scaling relations to calculate the

source magnitude, is another aleatory term. Although homogenous slip distribution is a common implementation in PTHA, slip distribution of large events do not show homogenous behavior. Therefore, definition of asperities on the fault is another aleatoric variable which should be considered. Tsunami numerical modelling, itself, is also another aleatoric variable since, they do not show correlation with real observations, which are more variable than earthquake scenarios incorporated in PTHA

(Grezio et al., 2017). The aleatory variable affects the results because it is incorporated directly into the hazard calculations (Abrahamson and Bommer, 2005).

Epistemic uncertainty, on the other hand, consists of the lack of knowledge of the physical process and data. Segmentation of fault system is one of the epistemic variables since, it is not certain where the rupture will be generated, and which segments will be triggered. In addition, there are many different scaling relations, which cause another epistemic uncertainty, between

the fault area and magnitude. It is also important for tsunami generation whether the fault rupture reaches to the surface or not. Thus, updip and downdip limits of the fault rupture can be considered as another epistemic variable (Grezio et al.,2017). Accurate probability distributions of input cannot be known. For example, assuming that probability of occurrence of an event follows Poisson distribution. However, return periods of events do not simply fit to this distribution (Gonzalez et al., 2013). Unlike aleatoric one, epistemic uncertainty can be decreased when more information is available (Godinho, 2007). Different

techniques, such as logic tree, Bayesian method etc., have been developed to reduce these uncertainties.

In this study, probabilistic model is established based on the characteristic fault model of PIF, which is a segment of NAF one of the best studied fault zones in the world. It is also assumed that the entire fault area is ruptured, reached to the surface and generated a homogenous slip for each event. The maximum magnitude range of the fault is calculated with Wells and Coppersmith (1994) scaling relation. All these assumptions naturally include uncertainties which are naturally reflected to this

PTHA study. Besides, MC simulation itself also includes uncertainty as being performed hundred times to create synthetic earthquake scenarios. The effect of uncertainty in aperiodicity parameter is also existing and can be reduced by including different parameters to MC simulation. Therefore, the tsunami hydrodynamic parameters associated with the probability of occurrence of the corresponding scenario preserve the same uncertainty.

**5 Conclusion**

In this study, time-dependent PTHA is performed in Tuzla region, Istanbul for the purpose of understanding the probability of having tsunami inundation after the PIF rupture. The study combines tsunami numerical modelling with probabilistic approach, which is modified by probabilistic seismic hazard analysis. Probability calculations have been done based on the time-dependent BPT model, which depends on the time period passed since the last characteristic event and the recurrence time of earthquake. After that, synthetic earthquake catalogue is generated using MC simulation technique and tsunami numerical

modelling was performed depending on this earthquake catalogue using NAMI DANCE code in GPU environment.

Results of this PTHA study was presented in three different ways for the next 50 and 100 years. The first one was the graphs showing the change of probability with the maximum and minimum water surface elevation and inundation depth for different time intervals. Secondly the probabilistic tsunami inundation maps are generated for Tuzla region. Finally, the probability

maps of exceedance of 0.3 m wave heights at synthetic gauge points are represented with bar charts.

The main results of this study can be summarized as follows:

- According to the distribution of probability with respect to tsunami hydrodynamic parameters, the probability of exceedance of 1 m maximum positive and negative water surface elevation is 65% within next 50 years. The probability for 1 m inundation depth is up to 60%.

- Considering probabilities for next 100 years, 85% probability of exceedance of 1 m was calculated. For 1 m inundation depth, probability of exceedance is obtained about 80%.

- As a result of the whole simulation, 1.8 m, -2.1 m and 1.6 m were calculated for maximum and minimum water surface elevation and inundation depth, respectively with the probability of 35% for the next 50 years, 60% for next 100 years.

- Inundation maps, indicate that inundation of tsunami waves that are equal to or larger than 0.3 m have probability mostly higher than 10 % and 20% for the next 50 years and 100 years, respectively. The probability of occurrence of 0.3m inundation depth was calculated as maximum 75% for the next 100 years. Maximum inundation distance is calculated as 60 m and observed in the southern part of the finest 3m grid-sized study area.

- Probabilistic results for the exceedance of 0.3 m wave height at synthetic gauge points demonstrate that only few of
them have a probability between 75% - 85%, however several points have more than 90% probability for the next 50 years. Probability of exceedance increases by more than 95% for the next 100-years.

The tsunami impact of the PIF rupture along the Tuzla coast is very important as proposed with the results of this study. However, as further steps of this study, PTHA can be done for the other critical test sites along the Marmara Sea that are close to the PIF segment. Besides, it is also advantageous to consider the other fault segments, having their various rupture
combinations and complex rupture probabilities in Marmara Sea as further studies. Previously in the framework of the MARSite project, tsunami arrival times and maximum wave amplitudes are calculated along the coast of the Marmara Sea using different earthquake scenarios and a tsunami scenario database was obtained in deterministic approach (Ozer Sozdinler et al., 2019). Results of this study show that, arrival time of tsunami waves is very short in Marmara Sea for most of the scenarios which complicates the tsunami early warning operations and evacuation actions. However, due to the short arrival
times of first tsunami waves along Marmara coast, the tsunami inundation scenario databases would be of great importance in such conditions. It would be the best option for the decision makers and civil protection authorities to have the inundation maps prepared also in probabilistic approach in order to realize the possibility of exceedance of selected threshold inundation depth for certain critical coastal locations.

This study shows a methodology for PTHA with time – dependent probabilistic model using only one fault (PIF) as earthquake
and tsunami source. Furthermore, this study can be developed including some connected faults to the PIF in both time – dependent and time - independent probability calculations and Brownian passage time (BPT) probability can be combined with static Coulomb stress changes on the faults. Brownian passage time (BPT) model can also be improved by including different aperiodicity parameters. The probability of occurrence of earthquakes is the main focus of this study to perform

tsunami hazard analyses. However, submarine landslides are other critical important sources for tsunami generation in

Marmara Sea. Probabilities of sliding areas and the sliding volumes can be considered in the analyses. Submarine landslide

generated tsunamis can be coupled with the earthquake triggered tsunamis in order to obtain integrated PTHA in the Marmara

Sea.

### Acknowledgements

The authors would like to acknowledge the project MARsite - New Directions in Seismic Hazard assessment through Focused

Earth Observation in the Marmara Supersite (FP7-ENV.2012 6.4-2, Grant 308417) and SATREPS Project MaRDiM

(Earthquake and Tsunami Disaster Mitigation in the Marmara Region and Disaster Education in Turkey). The authors would

like to especially thank Prof. Dr. Ahmet Cevdet Yalçıner for his valuable feedback and effort during this study and great

support of Bora Yalçıner and Andrey Zaitsev in the development and improvement of the NAMI DANCE numerical code. We

also would like to thank Dr. Öcal Necmioğlu, Dr. Maura Murru, Dr. Giuseppe Falcone, Prof. Dr. Semih Ergintav, Prof.

Dr. Sinan Akkar, Dr. Mine Demircioğlu and Prof. Dr. Mustafa Erdik for their valuable support and feedback. The Generic

Mapping Tools (GMT; Wessel and Smith, 1998) was used for plotting tectonic map of Turkey and bathymetric map of

Marmara fault system. Other maps throughout this paper were created using ArcGIS® software by Esri. ArcGIS® and

ArcMap™ are the intellectual property of Esri and are used herein under license. Copyright © Esri. All rights reserved. For

more information about Esri® software, please visit www.esri.com.

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
