# Peer review of "Probabilistic Tsunami Hazard Analysis for Tuzla Test Site Using Monte Carlo Simulations"

_Natural Hazards and Earth System Sciences, 2019_

## Referee Comment (RC1) · Anonymous Referee #1 · 4 Nov 2019

Dear Editor, Please find below the comments on the paper "Probabilistic Tsunami Hazard Analysis For Tuzla Test Site Using Monte Carlo Simulations" by H. Basak Bayraktar and Ceren Ozer Sozdinler.

General Comments The paper is interesting for the Tuzla test site and it deserves publication with some revisions. Major revisions should be focused on the discussion of the uncertainties: the length of the synthetic earthquake catalogue and the choices of the parameters. Are 100 earthquakes enough to cover a wide range of scenarios necessary for such a detailed probabilistic analysis for the Tuzla test site? What are we missing? A part from the definitions of the aleatoric and epistemic uncertainties, section

4.4 should present a deeper discussion. Also see the paper "Quantification of source uncertainties in Seismic Probabilistic Tsunami Hazard Analysis (SPTHA), by J. Selva, R. Tonini, I. Molinari, M.M. Tiberti, F. Romano, A. Grezio, D. Melini, A. Piatanesi, R. Basili and S. Lorito, Geophys. J. Int. (2016) 205, 1780–1803, doi: 10.1093/gji/ggw107 ". Figures 5-6 and Figures 7,8,9,10 present unreadable legends and labels, the authors should use a bigger font. Figure 11 (second panel) has undistinguishable colours for the bars, they should be changed.

Detailed Comments - At line 12 please insert "(PIF)" after "Prince Island Fault". - At line 13 please write "moment magnitudes" instead "magnitudes". - Please re-write the sentence from line 30 to line 34 because it is too long and it is not clear the meaning. - Please remove "in 2004" at line 34. - At line 39 it is wrong the use of the word "attractive" in this contest, please change or explain. - At line 40 remove "of" before large. - At line 79 is it "Mw>7" instead of "M>7" ? - At line 87 not clear how the small faults generate the tsunami. It is understandable for the submarine failures. Please explain. - At line 125 please remove "In tsunami research, this method" and write "It has been applied..." - At line 125 "Grezio et al. 2017" is not in the references. - At line 126 please write "the method is generally adapted" inserting the verb "is". - At line 157 maybe it is "was defined", is it missing the verb "was"? - Please keep the acronyms in the text MC (Monte Carlo) and PIF (Prince Island Fault). - Please remove the title of the subsection "2.1 Probability Calculations" as well as the lines 213-214 of the first sentence. - Line 226: the sentence "Time dependent probabilistic model is followed for the probability calculations; because , instead of using multi – segment rupture scenarios, only one fault is considered. " is not clear, please explain it. - It is better to write parameters and variable using the subscribed mode, for example Tr, Mw, Mo, and so on, because in formula (3) the "$2Tr\alpha2t$" seems to have 4 variables and not 3. - At line 258 please remove "in the next". - At line 307 the following sentence should be re-written: "First, graphics are prepared to show general distribution of probability of occurrence with respect to considered tsunami hydrodynamic parameters, which are minimum and maximum water surface elevation and inundation depth". A possible suggestion is the following:

"First, distribution of probability of occurrence of the tsunami hydrodynamic parameters, which are minimum and maximum water surface elevation and inundation depth, are shown". - At line 312 please change the title of the subsection simply by removing the words "Graphs of". - Lines 320-322 in Figure 5, graphics of probabilities of occurrences according to maximum and minimum water surface elevation (maximum water withdraw) and inundation depth for next 50 years are represented, respectively. According to these graphs, tsunami wave heights up to 1 m and withdrawal of the waves around 1 m have approximately 65% probability of occurrence. - At line 338 please remove "from the graphics". - Please re-write lines 347-349, they are not clear them. If I understand well your simulation of the worst earthquake case scenario produced the maximum water surface elevation equal to 1.85 m, the minimum water surface elevation (maximum withdraw) equal to 2.16 m and the inundation depth of 4.48 m and the probability of this worst earthquake case scenario is 35% for next 50 years and 60% for next 100 years. In the main text of the paper the residual are not mentioned, please write an explanation there (not only in the captions). - It is better to indicate the section 4.3 simply writing "Synthetic Gauges" and write in the text the approximate average distance between the points. - At line 436 the ";" should be ":". - At line 454 "from" should be changed in "by". - Lines 460-464 should be re-written, not clear what the authors intend by "results of the numerical modelling was demonstrated", "demonstration of results" and "finale outcomes".

Figures - Figure 1 is small and the legend is difficult to read. I suggest to use landscape for Figure 1 and to enlarge the legend. Please provide indication for the orange colour dots. - Figure 5 and 6 are difficult to understand, the font of the legend is too small and the red writing cannot be read. - Figure 11 (second panel) can improve the reading using the colour blue or violet for the bar instead of the red.

---

## Author Comment (AC1) · 8 Nov 2019

**RESPONSE TO REFEREE#1**

1) Major revisions should be focused on the discussion of the uncertainties: the length of the synthetic earthquake catalogue and the choices of the parameters. Are 100 earthquakes enough to cover a wide range of scenarios necessary for such a detailed probabilistic analysis for the Tuzla test site? What are we missing? A part from the definitions of the aleatoric and epistemic uncertainties, section 4.4 should present a deeper discussion.

Uncertainties section was re-written in detail and it can be explained ,why only 100 earthquakes are used, with :" NAFZ generates an earthquake with the recurrence interval of about 250 years beneath the Marmara Sea. Therefore, selecting 100 earthquake scenarios would cover a time period of 100x250 years= 25,000 years which is considered as an adequate catalog duration in this study. However, because of having time dependent probabilistic analyses, this catalog duration is not used for PTHA in this study."

2)Please re-write the sentence from line 30 to line 34 because it is too long and it is not clear the meaning.

This part was changed with: "The Marmara Sea and the area is one of the most seismically active areas in Turkey. Main active faults of the region pass through under the Marmara Sea. Thus, coastal cities in Marmara region, especially Istanbul which has significant importance in terms of economy, and historical and sociocultural heritage with a population of more than 15 million, is under the threat of high damage due to possible big earthquake and also triggered tsunamis."

3)At line 39 it is wrong the use of the word "attractive" in this contest, please change or explain.

The region has distinctive characteristics in terms of its complex tectonic structure and high possibility of an earthquake occurrence with the magnitude larger than 7.0 offshore Istanbul mega-city.

4)At line 79 is it "Mw>7" instead of "M>7" ?

Magnitude type of the expected event is not specified in the reference paper (Ergintav et al.,2014).

5)At line 87 not clear how the small faults generate the tsunami. It is understandable for the submarine failures. Please explain.

In this sentence, several small faults past was chanced with:" E -W trending tectonic deformation along the basin"

6)At line 125 "Grezio et al. 2017" is not in the references.

Paper was added to the reference list.

7)Please keep the acronyms in the text MC (Monte Carlo) and PIF (Prince Island Fault).

Text was modified changing Monte Carlo and Prince Island Fault with their acronyms.

8) Line 226: the sentence "Time dependent probabilistic model is followed for the probability calculations; because , instead of using multi – segment rupture scenarios, only one fault is considered. " is not clear, please explain it.

This sentence was changed with :" Time dependent probabilistic model is followed for the probability calculations; because, this probabilistic model allows to consider only one fault instead of using multi – segment rupture scenarios through characteristic earthquake model."

9)It is better to write parameters and variable using the subscribed mode, for example Tr, Mw, Mo, and so on, because in formula (3) the "2Tr_2t" seems to have 4 variables and not 3.

All the parameters were replaced with subscribed modes in the formulations and text.

10)At line 307 the following sentence should be re-written: "First, graphics are prepared to show general distribution of probability of occurrence with respect to considered tsunami hydrodynamic parameters, which are minimum and maximum water surface elevation and inundation depth". A possible suggestion is the following: "First, distribution of probability of occurrence of the tsunami hydrodynamic parameters, which are minimum and maximum water surface elevation and inundation depth, are shown".

This part was changed as recommended.

11)Lines 320-322 in Figure 5, graphics of probabilities of occurrences according to maximum and minimum water surface elevation (maximum water withdraw) and inundation depth for next 50 years are represented, respectively. According to these graphs, tsunami wave heights up to 1 m and withdrawal of the waves around 1 m have approximately 65% probability of occurrence.

The comment regarding this sentence is not clear.

12)Please re-write lines 347-349, they are not clear them. If I understand well your simulation of the worst earthquake case scenario produced the maximum water surface elevation equal to 1.85 m, the minimum water surface elevation (maximum withdraw) equal to 2.16 m and the inundation depth of 4.48 m and the probability of this worst earthquake case scenario is 35% for next 50 years and 60% for next 100 years. In the main text of the paper the residual are not mentioned, please write an explanation there (not only in the captions).

This paragraph was written clearer: "Considering the results of the whole simulations, the worst case earthquake scenario generated tsunami waves with maximum water surface elevation is equal to 1.8 m, minimum water surface elevation (maximum withdraw) is equal to 2.1 m and inundation depth is equal to 1.6 m. The probability of occurrence of this event is 35% for next 50 years and 60% for next 100 years."

13)Lines 460-464 should be re-written, not clear what the authors intend by "results of the numerical modelling was demonstrated", "demonstration of results" and "finale outcomes".

These conclusion remarks are re-written: "Results of this PTHA study was presented in three different ways for the next 50 and 100 years. The first one was the graphs showing the change of probability with the maximum and minimum water surface elevation and inundation depth for different time intervals. Secondly the probabilistic tsunami inundation maps are generated for Tuzla region. Finally, the probability maps of exceedance of 0.3 m wave heights at synthetic gauge points are represented with bar charts."

14)Figures - Figure 1 is small and the legend is difficult to read. I suggest to use landscape for Figure 1 and to enlarge the legend. Please provide indication for the orange colour dots. - Figure 5 and 6 are difficult to understand, the font of the legend is too small and the red writing cannot be read. - Figure 11 (second panel) can improve the reading using the colour blue or violet for the bar instead of the red.

All the figures are modified regarding to referee comments.

[revised manuscript text omitted]
, because it considers all possible earthquakes in a fault even they occur with very low probability and when decision makers design coastal zones and structures, especially critical ones, they would consider the results of probabilistic studies. Different from previous probabilistic approaches in Marmara Sea, in this study tsunami hazard assessment is done in the light of high possibility of occurrence of a big earthquake in Marmara Sea in the case of PIF rupture. The probability of earthquake occurrences in PIF are taken into account for the preparation of high-resolution tsunami inundation maps and distribution of hydrodynamic parameters due to the probability of occurrence of associated earthquakes on PIF determined by MC Simulations.

[revised manuscript text omitted]

MC simulation technique is generally applied to generate earthquake catalogue of a given length of time. In this technique, a list of earthquakes can be generated using the frequency - magnitude relationship for each seismic source (Zolfaghari, 2015). Seismic zonation should be done by considering regions that have relatively homogeneous earthquake activity and faulting regimes (Sørensen et al., 2012). After that, tsunami numerical modelling is performed for each event of this synthetic catalogue and tsunami hydrodynamic parameters, mainly maximum wave heights, inundation depth, current velocities, as well as tsunami inundation zones are estimated. Regional PTHA studies can be used as a guide for further local studies to develop of a more effective tsunami warning system. 
[revised manuscript text omitted]

---

## Referee Comment (RC2) · Anonymous Referee #2 · 29 Nov 2019

The authors have conducted a Probabilistic Tsunami Hazard Analysis (PTHA) of the strategic Tuzla area using Monte Carlo simulations. The work arrives at a $90\%/95\%$ probability of exceedance for $0.3m$ wave height for the next $50/100$ years. As an important ingredient for future hazard planning, this and the other results in the paper, showcase the relevance of such PTHA studies. As the authors rightly highlight in the conclusions, the PIF is very close to Tuzla and hence the tsunami arrival times are short. This further accentuates the importance of the hazard maps generated in this work for planners.

**General comments**

[Figure]

1. The paper needs major revisions to increase its readability and clarify the adopted methodology (*esp.* the MC simulations). The title of the paper contains the word *"Monte Carlo simulations"*. Hence, it is pertinent that the authors include a succint description of what they mean by the term and how they actually go about utilizing the MC methodology in the context of the paper. For example, any picture showing their sampling of the different parameters (magnitude and depth) would be helpful to the readers to visualize the different scenarios. Thus, in this regard, the current description in Section 2 is inadequate.

2. While generating the $100$ scenarios for the MC simulations, the PIF fault is defined as a characterisitic fault. This crops up in many places in the manuscript. Is this choice because of computational constraints with the simulations or difficulties in applying the MC methodology to multi-segment ruptures or does the seismicity in the PIF does not warrant it? It will be helpful if the authors elaborate on this aspect/choice.

**Specific comments on the technical content**

1. Throughout the paper the tsunami wave height, inundation depth *etc.* are mentioned as hydrodynamic *"parameters"*. Since the term *parameter* is usually used for physical constants in a model, or for independent variables, it is suggested that another word (*e.g. quantities*) may be used instead.
2. Lines 38-39 – Inclusion of references for studies *"regarding the fault mechanisms, ... and triggered tsunamis"* will benefit the readers.
3. Lines 39-40 – Please consider merging or rephrasing the sentence *"The region is attractive ... mega-city."* as it seems to be a repetition of facts in lines 34-37.
4. Line 48 – Please expand the abbreviation PIF the first time it occurs.
5. Lines 44-45 – Figure 1 gives a nice overview of the seismicity in the region. In the inset figure inside Figure 1, the labels describing the general tectonic map of Turkey are not clear even after zooming in. Consider either enlarging the inset figure or increasing the fonts of the labels or increasing resolution of the image.

[Figure]

6. Lines 59-61 – It is not clear why the sentence *"Therefore, making ... quite difficult"* has been added. Does this difficulty somehow influence the methodology or modelling in the paper?

7. Lines 75-78 – Consider splitting the sentence *"After the 1999 Izmit ... megacity Istanbul"* for increasing readability and clarifying the flow of thought.

8. Lines 78-90 – The critical importance of the PIF fault in generating the next earthquake has been brought out nicely in this paragraph. Inclusion of concrete numbers/facts from Ergintav et al. will add strength to the argument.

9. Line 82 – Please list suitable references related to *"studies on historical tsunami records"* at the end of the sentence. Would it be possible to replace *"majority"* by a concrete number?

10. Line 86-88 – A restructuring of the sentence *"The recent one ... tsunami."*, would make this fact more readable.

11. Line 91 – Please list suitable references related to *"tsunami hazard estimation studies"* at the end of the sentence.

12. Lines 93-95 – A rewording of the sentence *"When focused on ... normal component"*, would make the argument clearer.

13. Lines 98 – Please list suitable references related to *"probabilistic seismic and tsunami hazard analyses"* at the end of the sentence.

14. Lines 105-107 – Consider restructuring the sentence *"However, probabilistic ... probabilistic studies."*, for better clarity. The reades will benefit more, if the authors can cite a few other reasons as to why a probabilistic hazard assessment is important.

15. Lines 107-109 – The difference of the current work from previous approaches needs to be made clearer here. I found it difficult to lock on to the unique contribution of the paper by reading this.

16. Line 128 – Please give the expansion of the abbreviation *"SPTHA"*.

17. Lines 127-131 – The sentence *"Such studies ... maps"* is quite long. Consider splitting it up for clearing the flow of the argument.

18. Line 137 – Please give the expansion of the abbreviation *"PSHA"*. The word

*"should"* is quite strong. Consider replacing it by a milder alternative or adding a sentence or two to justify its usage.

19. Line 139 – Please clarify the phrase *"passive margins"*.

20. Line 144 – Please list suitable references related to *"Paleoseismologic studies"* here.

21. Line 146-147 – It is not at all clear what the authors intend to convey by the sentence *"According to Aki ... seismic cycles"*.

22. Line 170 – The authors are requested to clarify if they perform *"seismic zonation"* for this work, or use existing results.

23. Lines 174-175 – The sentence *"Tsunami risk ... local ones"*, seems to be a repetition of the previous one. Please consider merging the two sentences.

24. Line 180 – The term *"randomly"* is quite vague. The authors are requested to situate the term in their context by giving more details about say, the sampling *etc.*.

25. Line 182 – It is not clear what is conveyed by *"represents the number of iterations randomly done in MC simulations"*. The authors can either explain it/cite references as to why $100$ seems to be representative of the number of samples.

26. Line 192 – The authors are requested to verify the range of magnitudes *"Mw 6.5-7.1"* vis-a-vis the constants given in line 194. It would be better to supply the particular values of $a$ and $b$ (apart from the standard errors and deviations) that have been used to arrive at the range.

27. Lines 206-211 – The authors are requested to clarify what is the range of depths used for generating the scenarios. Is it $5 - 14\,km$ or $10 - 14\,km$? It would be beneficial to the readers if the ranges of parameters used for the MC simulations were included in Table 1, alongside the other fault parameters.

28. Line 218 – Please list suitable references related to *"Brownian passage time (BPT), log-normal or another probability distribution"* here.

29. Lines 218-220 – The sentence *"In this model ... elapsed time."*, needs to be rephrased or split for better readability.

30. Lines 221-224 – It is not clear what the authors mean to convey by this paragraph.

Is this relevant to the application of the method used in this work?

31. Lines 228-232 – This paragraph is a repetition of lines 70-74 and should be deleted.

32. Line 236 – The authors can share the reasons for adopting the simplification: *"earthquake releases all energy loaded on the fault and then starts the new failure cycle."*. Is it because of lack of earthquake cycling models with residual energy or perhaps, due to incompatibility with the BPT model?

33. Lines 247-249 – The authors are suggested to cite a reference for the definition of $Tr$ so that interested readers may look into it.

34. Lines 269-270 – The linear version of the SW equations is usually faster than the non-linear version, needs lesser memory and is accurate in deep water where non-linear effects may be neglected. Thus, the reasons given by the authors here are not convincing. A NSW model is, of course, a better model than the linear case because it models the physics better. The authors can supply a sentence or two as to why the use of NSW is attractive in this work.

35. Line 274 – THe authors have used Okada equations for calculating deformation due to the fault. A figure of the PIF segmentation (maybe in Figure 2, left figure) portraying one of the scenarios from the 100 cases would be helpful to visualize the fault.

36. Lines 279-280 – Accurate coastal bathymetry is crucial for accuracy in high-resolution simulations near the coast. The authors can clarify the source for the bathymetry-topography data at $3m$ resolution. Is it simply the downscaled version of $30m$ ASTER and $900m$ GEBCO data? Or is it another, local dataset? Also, when merging the different bathymetry datasets, a common problem is the fixing of the coastline. How do the authors decide the coastlines between GEBCO and ASTER, as well as the digitized coastline from ArcGIS(?)? A few sentences describing their adopted methodology would be appropriate.

37. Line 292 & Figure 4 – The ratio $1:3$ has been employed here for scaling of consecutive mesh resolutions for simulation. Is there a similar guideline for appropriate spacing between the different nested grids? In Figure 4, the nesting rectangles for $3m$, $9m$, and $27m$ are bunched together on their right (eastern) sides. This would create

a sharp gradient of mesh resolution from $3m$ to $27m$ in that region. The authors can comment on the possible repurcussions due to this on the results.

38. Lines 323-325 – The authors make a good point about the importance of the minimum wave height in terms of stranding of ships. A citation/past example would make this point even better.

39. Lines 348,471 – The inundation depth calculated in worst case earthquake scenario is given as *"$4.48\,m$"*. I am unable to find this number in Figures 5 or 6. The authors can clarify this. Also, should it be *"$2.16\,m$"* or $-2.16\,m$, as is written in line 471?

40. Line 352 – I was very interested in the hazard of small amplitude waves dragging people. A citation would be relevant for readers (like me) who would want to dig more into this aspect.

41. Line 363 - Do the authors mean *"mean (average) inundation depth values"* or *mean (average) probability values*? It would be more clarifying to include a step-by-step/point-by-point procedure of calculating the excedance probablities from the 100 MC simulations.

42. Lines 434-444 - This discussion of uncertainties does not shed enough light on the results. As such, it is better positioned in the methodology section (maybe? before line 233/discussion of BPT model). Otherwise, the authors can expand this section a bit more with concrete connections to the numbers in the numerical results.

**Technical corrections**

The introduction (*i.e.* till line 116) is relatively free of typos. However, the authors are recommended to check the grammar and sentence constructions in the rest of the paper. I list below (a few) corrections for the typos that I have come upon. The authors need not follow these corrections exactly, and can make their modifications, as long as readability is maintained.

1. Line 64 – Replace *"ranging"* with "ranges".
2. Lines 37,75,79 – Replace *"$M > 7$"* with *"$M_w > 7$"*.
3. Line 76 – Insert comma after *"event"* . Replace *"extend"* with *"extends"*.

4. Line 82 – Insert *"The"* before *"1509"*.

5. Line 85 – Insert *"The"* before *"1894"*.

6. Line 89 – While using multiple ands, a comma seems appropriate, especially after *"Hereke"*.

7. Line 98 – Replace *"are"* by *"were"*.

8. Lines 125-126 – Replace *"source, such as earthquakes, landslide, volcanic activity etc. in various scales,"* by *"sources, such as earthquakes, landslides, volcanic activities etc. at various scales;"*.

9. Line 127 – Insert *"is"* after *"method"*.

10. Line 128 – Insert *"are"* before *"generated"*.

11. Line 138 – Replace *"primarily"* by *"primary"*.

12. Line 141 – Insert *"are"* before *"exposed"*.

13. Line 150 – Replace *"constant"* by *"constants"*.

14. Line 151 – Replace *"package"* by *"packages"*.

15. Line 157 – Replace *"segments"* by *"segment is"*.

16. Line 158 – Replace *"PIF, as given"* by *"PIF are given"*.

17. Line 159 – Delete *"the"* before *"Armijo"*.

18. Line 160 – Delete *"is"* before *"also fits"*. Replace *"constant"* by *"constants"*.

19. Line 174 – Delete *"of"* before *"a more"*.

20. Line 181 – Replace *"for to having"* by either *"for having"* or *"by having"*, as appropriate.

21. Line 194 – Replace *"errors"* by *"error"*.

22. Line 205 – Replace *"scenarios"* by *"scenario"*.

23. Line 213 – Replace *"is describing"* by *"describes"*.

24. Line 216-217 – Insert *"is*" after *"occurence"*. Replace *"that was passed"* with *"that has passed"*.

25. Line 219 – Consider rephrasing *"... the additional required information in addition to ..."*.

26. Line 258 – Delete repeated phrase *"in the next"*.

27. Line 303 – Insert *"years* after *"and 100"*.
28. Line 324 – Replace *"is important as much as of"* by *"is as important as of"*.
29. Line 434 – Replace *"includes"* by *include*. Replace *"rare occurence nature"* by *"nature of rare occurence"*.
30. Line 437 – Replace *"effects"* by *"affects"*.
31. Line 468 – Consider deleting the word *"decreases"*, as it appears misleading here.
32. Line 479 – Replace *"them has a"* with *"them have a"*.

---

## Author Comment (AC2) · 19 Dec 2019

**Response to Referee #2**

**General comments**

1. The paper needs major revisions to increase its readability and clarify the adopted methodology (esp. the MC simulations). The title of the paper contains the word "Monte Carlo simulations". Hence, it is pertinent that the authors include a succint description of what they mean by the term and how they actually go about utilizing the MC methodology in the context of the paper. For example, any picture showing their sampling of the different parameters (magnitude and depth) would be helpful to the readers to visualize the different scenarios. Thus, in this regard, the current description in Section 2 is inadequate.

Authors appreciate this comment. In fact, explanations for the application of MC simulations have already been revised in Section 2 according to the first referee's comments. We kindly request the referee to reread especially the lines between 175-195 for rephrased explanations.

2. While generating the 100 scenarios for the MC simulations, the PIF fault is defined as a characterisitic fault. This crops up in many places in the manuscript. Is this choice because of computational constraints with the simulations or difficulties in applying the MC methodology to multi-segment ruptures or does the seismicity in the PIF does not warrant it? It will be helpful if the authors elaborate on this aspect/choice.

Application of MC simulation technique and the calculation of multi-segment rupture probabilities is a complex problem. Therefore, in this study, only single fault is selected as a source and characteristic model is applied to PTHA as a basic case study. This study can be developed including some connected fault segments to the PIF with static Column stress changes, as mentioned in the manuscript.

**Specific comments on the technical content**

1. Throughout the paper the tsunami wave height, inundation depth etc. are mentioned as hydrodynamic "parameters". Since the term parameter is usually used for physical constants in a model, or for independent variables, it is suggested that another word (e.g. quantities) may be used instead.

In literature, the general classification of wave parameters such as velocity, acceleration, height etc. are "hydrodynamic parameters". Also, there are many examples of using the phrase "tsunami hydrodynamic parameters" in literature (i.e. Brill et al., 2014https://www.jcronline.org/doi/abs/10.2112/JCOASTRES-D-13-00206.1?journalCode=coas). Therefore, authors would prefer using the term "parameter".

2. Lines 38-39 – Inclusion of references for studies "regarding the fault mechanisms, ... and triggered tsunamis" will benefit the readers.

Following references are added to the manuscript; Armijo et al., 2002; Armijo et al., 2005; Okay et al., 1999; Le Pichon et al., 2001; Yaltirak 2002; McNeill et al., 2004; Aksu et al., 2000; Imren et al., 2001; Le Pichon et al., 2001; Pondard et al., 2007, Yalçıner et al., 1999; Yalçıner et al., 2002; Aytore et al., 2016 ; Hebert et al., 2005; Altınok et al., 2001; Altınok et al., 2003; Guler et al., 2015; Cankaya et al., 2016; Tufekci et al., 2018; Latcharote et al., 2016

3. Lines 39-40 – Please consider merging or rephrasing the sentence "The region is attractive ... mega-city." as it seems to be a repetition of facts in lines 34-37.

The sentence was rephrased as: "The region has distinctive characteristics in terms of its complex tectonic structure and high possibility of an earthquake occurrence with the magnitude larger than 7.0 offshore Istanbul mega-city."

5. Lines 44-45 – Figure 1 gives a nice overview of the seismicity in the region. In the inset figure inside Figure 1, the labels describing the general tectonic map of Turkey are not clear even after zooming in. Consider either enlarging the inset figure or increasing the fonts of the labels or increasing resolution of the image.

The small figure that shows the tectonic mechanism of Turkey is enlarged.

6. Lines 59-61 – It is not clear why the sentence "Therefore, making ... quite difficult" has been added. Does this difficulty somehow influence the methodology or modelling in the paper?

Conducting a segmentation model for the offshore parts of the NAFZ is quite challenging, which causes the fault dimensions, such as its length and width, to include a sum of error margin. This command is also added to manuscript.

7. Lines 75-78 – Consider splitting the sentence "After the 1999 Izmit ... megacity Istanbul" for increasing readability and clarifying the flow of thought.

This sentence was split into two as, "After the 1999 Izmit event seismic energy along the 150 km long northern part of the NAFZ has been accumulated continuously since 22 May 1766 earthquake. This fault zone extend right next to south of Istanbul beneath the Marmara Sea, and this situation increases the rupture possibility of the PIF and the risk for megacity Istanbul (Stein et al., 1997; Barka 1999; Bohnhoff et al., 2013)."

8. Lines 78-90 – The critical importance of the PIF fault in generating the next earthquake has been brought out nicely in this paragraph. Inclusion of concrete numbers/ facts from Ergintav et al. will add strength to the argument.

Some certain numbers are included and the sentences was rephrased as, "Ergintav et al., (2014) also indicated that the PIF segment accumulates stress 15±2 mm/yr and the 3.7m slip deficit has been accumulated since the 1766 events and this makes PIF most likely to generate the next M > 7 earthquake along the Sea of Marmara segment of the NAF."

9. Line 82 – Please list suitable references related to "studies on historical tsunami records" at the end of the sentence. Would it be possible to replace "majority" by a concrete number?

In the literature, the certain number of the earthquake-related tsunami events is not mentioned. Therefore, authors prefer to use the word "majority".

10. Line 86-88 – A restructuring of the sentence "The recent one ... tsunami.", would make this fact more readable.

This sentence has been modified as "The recent one happened after the 17 August 1999 Izmit earthquake and after the earthquake, E-W trending tectonic deformation along the basin and submarine failures generated a tsunami."

11. Line 91 – Please list suitable references related to "tsunami hazard estimation

studies" at the end of the sentence.

Following references are added "Ozer Sozdinler et al., 2019; Hancilar, 2012; Aytore et al., 2016; Hebert et al., 2005"

12. Lines 93-95 – A rewording of the sentence "When focused on ... normal component", would make the argument clearer.

This sentence was rephrased as "The 40 km long fault in Eastern Basin of Marmara Sea, with a significant normal component, may generate tsunami wave which can reach maximum 2 m height along the Istanbul coast with locally considerable inundation (Hebert et al., 2005)."

13. Lines 98 – Please list suitable references related to "probabilistic seismic and tsunami hazard analyses" at the end of the sentence.

Following references are added; "Murru et al., 2016; Erdik et al., 2004; Hancilar, 2012"

14. Lines 105-107 – Consider restructuring the sentence "However, probabilistic ... probabilistic studies.", for better clarity. The reades will benefit more, if the authors can cite a few other reasons as to why a probabilistic hazard assessment is important.

The sentence was changed as "However, probabilistic tsunami hazard assessment is important to calculate the tsunami exposure and risk on human populations and infrastructures since probability calculations consider all possible earthquakes in a fault even they occur with very low probability (Lovholt et al., 2012; Lovhot et al., 2015; Grezio et al., 2017). "

15. Lines 107-109 – The difference of the current work from previous approaches needs to be made clearer here. I found it difficult to lock on to the unique contribution of the paper by reading this.

The statement was changed: "Different from previous probabilistic approaches in Marmara Sea, the probability of earthquake occurrences in one fault segment, PIF, are taken into account for the preparation of high-resolution tsunami inundation maps and distribution of hydrodynamic parameters due to the probability of occurrence of associated earthquakes on PIF determined by MC Simulations."

17. Lines 127-131 – The sentence "Such studies ... maps" is quite long. Consider splitting it up for clearing the flow of the argument.

This sentence was split into two; "Such kind of studies consider the events that generated by co-seismic sea floor displacement, Seismic Probabilistic Hazard Analysis (SPTHA), but numerous tsunami simulations are required to consider all expected combination of seismic sources. This problem can be solved by applying a simplified event tree approach and a two-stage filtering procedure to reduce the number of required source scenarios without decreasing the quality and accuracy of inundation maps (Lorito et al., 2015)."

18. Line 137 – Please give the expansion of the abbreviation "PSHA". The word "should" is quite strong. Consider replacing it by a milder alternative or adding a sentence or two to justify its usage.

The word "should be" was changed with "can be"

19. Line 139 – Please clarify the phrase "passive margins".

The explanation is added.

20. Line 144 – Please list suitable references related to "Paleoseismologic studies" here.

Following references are added: "Ryall et al., 1966; Allen, 1968; Schwartz and Coppersmith, 1984"

21. Line 146-147 – It is not at all clear what the authors intend to convey by the sentence "According to Aki ... seismic cycles".

The sentence is rephrased as "According to Aki (1984), characteristic earthquake is generated as a result of constancy of barriers to rupture through repeated seismic cycles."

22. Line 170 – The authors are requested to clarify if they perform "seismic zonation" for this work, or use existing results.

The following explanation is added: "In this study, fault segment model proposed in Ozer Sozdinler et al. (2019) is used and PIF is the only segment for seismic source."

23. Lines 174-175 – The sentence "Tsunami risk ... local ones", seems to be a repetition of the previous one. Please consider merging the two sentences.

The following sentence is removed: "Regional PTHA studies can be used as a guide for further local studies to develop
of a more effective tsunami warning systems.".

24. Line 180 – The term "randomly" is quite vague. The authors are requested to situate the term in their context by giving more details about say, the sampling etc..

The term "randomly" changed with depth as "uniformly distributed random numbers in a given interval"

25. Line 182 – It is not clear what is conveyed by "represents the number of iterations randomly done in MC simulations". The authors can either explain it/cite references as to why 100 seems to be representative of the number of samples.

This part was changed as "Therefore, selecting 100 earthquake scenarios would cover a time period of 100x250 years= 25,000 years which is considered as an adequate catalog duration in this study" regarding to comments of first referee.

26. Line 192 – The authors are requested to verify the range of magnitudes "Mw 6.5-7.1" vis-a-vis the constants given in line 194. It would be better to supply the particular values of a and b (apart from the standard errors and deviations) that have been used to arrive at the range.

The characteristic model assumes that an earthquake releases all of the seismic energy during the fault rupture and the magnitude of the earthquake depends on the dimension of fault (Abrahamson and Bommer, 2005) and PIF zone is assumed that it has potential to generate a characteristic earthquake and rupture the entire fault. Therefore, this fault can generate and event between the Mw 6.5 – 7.1 by rupturing entire fault area every time. Standard errors of a and b value are removed.

27. Lines 206-211 – The authors are requested to clarify what is the range of depths used for generating the scenarios. Is it 5 - 14 km or 10 - 14 km? It would be beneficial to the readers if the ranges of parameters used for the MC simulations were included in Table 1, alongside the other fault parameters.

This sentence was added "Therefore, depth of events vary between 5 to 14 km in MC simulations."

Authors prefer to keep Table 1 in this way, otherwise the information about depth and magnitude range will be given before the explanation of how they are retrieved.

28. Line 218 – Please list suitable references related to "Brownian passage time (BPT), log-normal or another probability distribution" here.

Following references are added "Matthews et al., 2002; Ellsworth et al., 1999;Davis et al., 1989; Rikitake 1974"

29. Lines 218-220 – The sentence "In this model ... elapsed time.", needs to be rephrased or split for better readability.
The sentence was rephrased as: "In this model, in addition to the recurrence time of earthquake, variability of the frequency of events and the elapsed time from the last characteristic event are the additional required information and the longer elapsed time causes to increase of probability of an event occurrence (Cramer et al., 2000; Petersen et al., 2007)."

30. Lines 221-224 – It is not clear what the authors mean to convey by this paragraph. Is this relevant to the application of the method used in this work?

The following sentence is added "Therefore, this model is suitable for calculating the probability of occurrence of an earthquake on a single fault."

31. Lines 228-232 – This paragraph is a repetition of lines 70-74 and should be deleted.

It was deleted.

32. Line 236 – The authors can share the reasons for adopting the simplification: "earthquake releases all energy loaded on the fault and then starts the new failure cycle.". Is it because of lack of earthquake cycling models with residual energy or perhaps, due to incompatibility with the BPT model?

The sentence is rephrased as: "A characteristic event occur when the load-state process reaches to the failure threshold; an earthquake releases all energy loaded on the fault and then starts the new failure cycle. "

33. Lines 247-249 – The authors are suggested to cite a reference for the definition of Tr so that interested readers may look into it.

The following reference is added "Ren and Zhang, 2013"

34. Lines 269-270 – The linear version of the SW equations is usually faster than the non-linear version, needs lesser memory and is accurate in deep water where nonlinear effects may be neglected. Thus, the reasons given by the authors here are not convincing. A NSW model is, of course, a better model than the linear case because it models the physics better. The authors can supply a sentence or two as to why the use of NSW is attractive in this work.

There is typo in this sentence, thanks the referee for the correction. It was corrected as Linear shallow water equations.

35. Line 274 – THe authors have used Okada equations for calculating deformation due to the fault. A figure of the PIF segmentation (maybe in Figure 2, left figure) portraying one of the scenarios from the 100 cases would be helpful to visualize the fault.

The figure is added to manuscript as Figure 5.

36. Lines 279-280 – Accurate coastal bathymetry is crucial for accuracy in highresolution simulations near the coast. The authors can clarify the source for the bathymetry-topography data at 3m resolution. Is it simply the downscaled version of 30m ASTER and 900m GEBCO data? Or is it another, local dataset? Also, when merging the different bathymetry datasets, a common problem is the fixing of the coastline. How do the authors decide the coastlines between GEBCO and ASTER, as well as the digitized coastline from ArcGIS(?)? A few sentences describing their adopted methodology would be appropriate.

The explanation for bathymetry- topography data was added.

37. Line 292 & Figure 4 – The ratio 1 : 3 has been employed here for scaling of consecutive mesh resolutions for simulation. Is there a similar guideline for appropriate spacing between the different nested grids? In Figure 4, the nesting rectangles for 3m, 9m, and 27m are bunched together on their right (eastern) sides. This would create a sharp gradient of mesh resolution from 3m to 27m in that region. The authors can comment on the possible repurcussions due to this on the results.

Although the nested domains seem like they are located very close to each other, this is just because of map resolution. In fact, there are enough number of grid nodes between two consecutive domains in order to have stable calculations in NAMIDANCE. Besides, it doesnt cause stability problem to have even only one grid node between two domains.

38. Lines 323-325 – The authors make a good point about the importance of the minimum wave height in terms of stranding of ships. A citation/past example would make this point even better.

The example of 20$^{th}$ July 2017 Bodrum-Kos earthquake and tsunami (Yalçıner et al., 2017) is added.

39. Lines 348,471 – The inundation depth calculated in worst case earthquake scenario is given as "4:48m". I am unable to find this number in Figures 5 or 6. The authors can clarify this. Also, should it be "2:16m" or -2:16m, as is written in line 471?

The inundation depth should be 1.6m instead of 4.48. Therefore, it was changed with 1.6m.

40. Line 352 – I was very interested in the hazard of small amplitude waves dragging people. A citation would be relevant for readers (like me) who would want to dig more into this aspect.

The following reference is added; "Jonkman and Penning-Rowsell, 2008"

41. Line 363 - Do the authors mean "mean (average) inundation depth values" or mean (average) probability values? It would be more clarifying to include a step-bystep/ point-by-point procedure of calculating the excedance probablities from the 100 MC simulations.

The calculations steps are explained gradually as suggested.

42. Lines 434-444 - This discussion of uncertainties does not shed enough light on the results. As such, it is better positioned in the methodology section (maybe? before line 233/discussion of BPT model). Otherwise, the authors can expand this section a bit more with concrete connections to the numbers in the numerical results.

The discussion of uncertainties has already extended according to comments of the first referee.

---

## Referee Report (RR1)

Dear Editor,
the paper "Probabilistic Tsunami Hazard Analysis For Tuzla Test Site Using Monte Carlo Simulations" by H. Basak Bayraktar and Ceren Ozer Sozdinler for the Tuzla test site improved after the changes made by the authors. The points listed by the reviewers were accomplished and the paper deserves publication.
In the last phase of the publishing procedure special attention should be made for the inclusion of all references.

Best Regards
Anita Grezio